# Penaeid shrimp genome provides insights into benthic adaptation and frequent molting

Xiaojun Zhang [1,2], Jianbo Yuan[1,2], Yamin Sun[3], Shihao Li[1,2,4], Yi Gao[1,2,4], Yang Yu[1,2,4], Chengzhang Liu[1,2,4], Quanchao Wang[1,2,4], Xinjia Lv[1,5], Xiaoxi Zhang[1,5], Ka Yan Ma[6], Xiaobo Wang[7], Wenchao Lin[4], Long Wang[4], Xueli Zhu[4], Chengsong Zhang[1,2,4], Jiquan Zhang[1,2,4], Songjun Jin[1,2,4], Kuijie Yu[1,2,4], Jie Kong[8], Peng Xu[9], Jack Chen[10], Hongbin Zhang[11], Patrick Sorgeloos[12], Amir Sagi[13], Acacia Alcivar-Warren[14], Zhanjiang Liu[15], Lei Wang[16], Jue Ruan[7], Ka Hou Chu[6], Bin Liu[16], Fuhua Li[1,2,4] & Jianhai Xiang [1,2,4]

Crustacea, the subphylum of Arthropoda which dominates the aquatic environment, is of major importance in ecology and fisheries. Here we report the genome sequence of the Pacific white shrimp *Litopenaeus vannamei*, covering ~1.66 Gb (scaffold N50 605.56 Kb) with 25,596 protein-coding genes and a high proportion of simple sequence repeats (>23.93%). The expansion of genes related to vision and locomotion is probably central to its benthic adaptation. Frequent molting of the shrimp may be explained by an intensified ecdysone signal pathway through gene expansion and positive selection. As an important aquaculture organism, *L. vannamei* has been subjected to high selection pressure during the past 30 years of breeding, and this has had a considerable impact on its genome. Decoding the *L. vannamei* genome not only provides an insight into the genetic underpinnings of specific biological processes, but also provides valuable information for enhancing crustacean aquaculture.

[1] CAS Key Laboratory of Experimental Marine Biology, Institute of Oceanology, Chinese Academy of Sciences, Qingdao 266071, China. [2] Laboratory for Marine Biology and Biotechnology, Qingdao National Laboratory for Marine Science and Technology, Qingdao 266237, China. [3] Tianjin Biochip Corporation, Tianjin 300457, China. [4] Center for Ocean Mega-Science, Chinese Academy of Sciences, Qingdao 266071, China. [5] University of Chinese Academy of Sciences, Beijing 100049, China. [6] School of Life Sciences, The Chinese University of Hong Kong, Shatin, N.T. 999077, Hong Kong SAR. [7] Agricultural Genomics Institute, Chinese Academy of Agricultural Sciences, Shenzhen 518120, China. [8] Key Laboratory for Sustainable Utilization of Marine Fisheries Resources of Ministry of Agriculture, Yellow Sea Fisheries Research Institute, Chinese Academy of Fishery Sciences, Qingdao 266071, China. [9] College of Ocean and Earth Sciences, Xiamen University, Xiamen 361102, China. [10] Department of Molecular Biology and Biochemistry, Simon Fraser University, Burnaby, BC V5A 1S6, Canada. [11] Department of Soil and Crop Sciences, Texas A&M University, College Station, TX 77843, USA. [12] Laboratory of Aquaculture & Artemia Reference Center, Ghent University, Coupure Links 653, Gent 9000, Belgium. [13] Department of Life Sciences and the National Institute for Biotechnology, Negev Ben Gurion University, Beer Sheva 84105, Israel. [14] Environmental Genomics Inc., P.O. Box 196 Southborough, MA 01772-1801, USA. [15] School of Fisheries, Aquaculture and Aquatic Sciences, Auburn University, Auburn, AL 36849, USA. [16] College of Life Sciences, Nankai University, Tianjin 300071, China. These authors contributed equally: Xiaojun Zhang, Jianbo Yuan, Yamin Sun, Shihao Li, Yi Gao, Yang Yu. Correspondence and requests for materials should be addressed to K.H.C. (email: kahouchu@cuhk.edu.hk) or to B.L. (email: liubin1981@nankai.edu.cn) or to F.L. (email: fhli@qdio.ac.cn) or to J.X. (email: jhxiang@qdio.ac.cn)

Crustacea is a dominant and diverse group of aquatic animals, among which the decapods (order Decapoda) comprise members of major ecological significance in marine and freshwater habitats and include many species with high economic values, such as shrimp, crabs, lobsters, and crayfish. In particular, the penaeid shrimp (family Penaeidae) represents the most important group in fisheries and aquaculture, and has therefore attracted considerable research attention[1].

Penaeids predominantly inhabit shallow seas in tropical and subtropical areas. Their life history typically involves migration between offshore and inshore waters. After spawning, the planktonic shrimp larvae in offshore waters develop into postlarvae, which move to inshore waters and become benthic. Thus the shrimps have to adapt to different environments in their life but the underlying biological mechanisms have seldom been explored. Similar to all arthropods, penaeids exhibit discontinuous growth through intermittent molting. Different from most insects, where molting usually occurs at the larval stages for growth and metamorphosis, molting occurs throughout the lifetime of crustaceans. For instance, the penaeid shrimp experiences about 50 molts during a lifetime[2], far more than in other arthropods[3]. Thus, shrimps could be an ideal model in studying the molting process. However, the detailed mechanisms of molting regulation are far from understood.

With an annual capture production of ~1.3 million tonnes, penaeids comprise the bulk of the global shrimp catch. Shrimp farming began in the 1970s, and has been one of the fastest growing sectors of the rapidly expanding aquaculture industry. The penaeid shrimp farming production reached >5 million tonnes, valued at over US$32 billion in 2016[4]. The Pacific white shrimp Litopenaeus vannamei is the most important cultured shrimp worldwide, accounting for ~80% of total cultured penaeid shrimp production. While its native range is in the East Pacific Ocean, L. vannamei is now widely farmed in Central and South America and in Asia, particularly in China, Indonesia, Thailand, and Vietnam, with a number of breeding lines available. These lines are mostly produced through traditional selective breeding procedures, and genomic information would be extremely useful for future genetic manipulation. As early as 1997, in an international workshop on genome mapping of aquaculture animals, the penaeid shrimp was identified as one of five target organisms for genome sequencing, together with salmon, catfish, tilapia, and oyster[5]. The genomes of the latter four species have been published over the past decade[6–9], but no complete genome of the shrimp has been reported to date, despite the efforts of a number of major research groups. For other crustaceans, high-quality genome assembly is only available for the water flea Daphnia pulex, the amphipod Parhyale hawaiensis, and the marbled crayfish Procambarus virginalis[10–12].

Here we present a high-quality de novo reference genome assembly for L. vannamei, with an extremely high proportion of simple sequence repeats (SSRs), which have been the major obstacle to genome sequencing and assembly. A prominent characteristic of the genome is the expansion of a series of genes related to visual system, nerve signal conduction, and locomotion, which apparently better equip the shrimp to adapt to its benthic habitat. The genome also sheds light on regulation of frequent molting through an intensified ecdysone signal pathway, with precise control by hormones and environmental factors. Moreover, our genomic analyses reveal that selective breeding has exerted a significant impact on the genome of L. vannamei broodstocks and the genetic resources acquired from this study will be useful for further genetic improvements in shrimp culture.

**Table 1 Summary of *L. vannamei* genome assembly**

| | |
|---|---|
| **Genome assembly statistics** | |
| Total length | 1,663,559,157 bp |
| Number of scaffolds | 4683 |
| Longest scaffold | 3,458,369 bp |
| Contig N50 length | 57,650 bp |
| Scaffold N50 length | 605,555 bp |
| Scaffold N90 length | 204,841 bp |
| **Genome characteristics** | |
| GC content | 35.68% |
| Content of transposable elements | 16.17% |
| Content of SSRs | 23.93% |
| Predicted protein-coding gene number | 25,596 |
| Predicted noncoding RNA gene number | 2777 |
| Quantity of scaffolds anchored on linkage groups | 3275 |
| Length of scaffolds anchored on linkage groups | 1,449,959,403 bp |

SSR simple sequence repeat, GC Guanine/Cytosine

## Results

**Genome sequencing assembly and annotation.** The *L. vannamei* genome size was measured to be 2.45 Gb by flow cytometry (Supplementary Fig. 1), similar to the size estimated by k-mer analysis (2.60 Gb, Supplementary Fig. 2). The sequencing and assembly of the *L. vannamei* genome were challenging due to the highly abundant SSRs as suggested by genome survey analysis[13]. We therefore applied a variety of sequencing technologies, which generated 828 Gb of Illumina clean sequences (338×), 133 Gb of PacBio long reads (54×), and 34,266 BAC end sequences (0.46×) (Supplementary Tables 1−3). We also conducted numerous conventional approaches for genome assembly, which yielded unsatisfactory results (Supplementary Table 4). Finally, we developed the WTDBG approach, which uses a fuzzy Bruijn graph method, to obtain the best assembly with the highest continuity and accuracy for the genome, that contains 1.66 Gb in 4683 scaffolds, with contig N50 of 57.65 Kb and scaffold N50 of 605.56 Kb (Table 1). The assembly was comparable to, or better than, those of other crustaceans, including the recently published genome of the marbled crayfish (scaffold N50 of 39.40 Kb)[10] (Supplementary Table 5).

The assembly showed high integrity and quality. Over 93% of Illumina reads could be mapped to the genome (Supplementary Table 6). The genome covered >94% of the unigenes assembled from RNA-Seq data and 94.76% of the conserved core eukaryotic genes (Supplementary Tables 7 and 8). The accuracy of our assembly is further substantiated by the 14 sequenced BAC clones, which were completely covered by corresponding scaffolds with high synteny (Supplementary Figs. 3 and 4). To assemble chromosome-level sequences, a high-density linkage map was used to anchor the scaffolds[13]. A total of 3275 scaffolds were anchored to 44 pseudochromosomes, representing 87.34% of the total genome (Fig. 1a, Supplementary Table 9).

The k-mer analysis indicated that repetitive sequences account for ~78% of the genome (Supplementary Fig. 2), which is more than that identified in the final assembly (49.38%) (Supplementary Table 10), indicating that some repetitive sequences are still missing from the assembly. The genome contains the highest proportion of SSRs (23.93%) among all sequenced animal genomes (Fig. 1b, Supplementary Table 11). The mean length of SSRs is 72.21 bp, which is over twice as long as those in other arthropods (20.11−31.91 bp). The SSRs density (3315.23/Mb, one SSR per 301 bp) in the *L. vannamei* genome is also, as far as we are aware, the highest among other reported genomes, except *Pediculus humanus* (4508.69/Mb) (Fig. 1b). Dinucleotide repeats are the dominant type of SSRs,

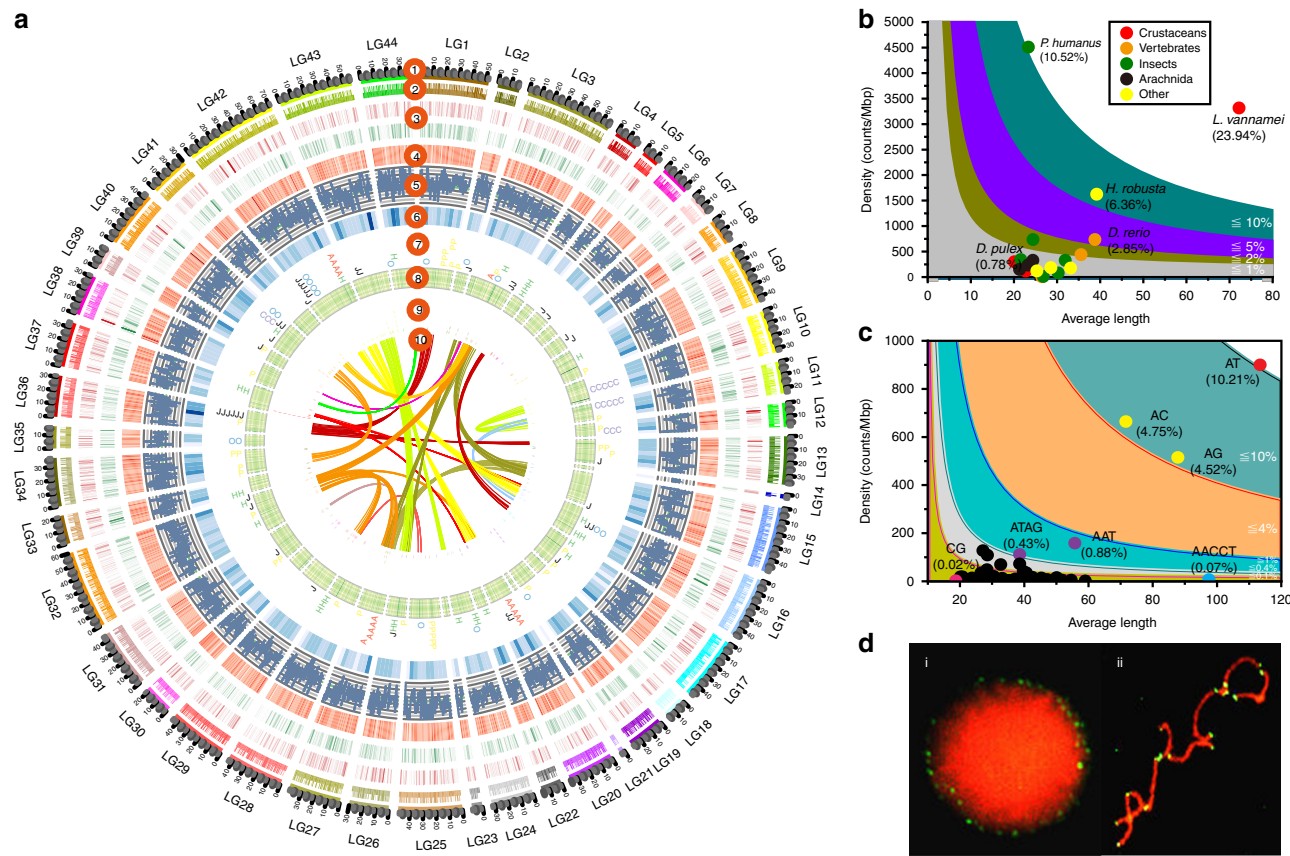

**Fig. 1** The genomic characteristics of *L. vannamei*. **a** A schematic representation of the genomic characteristics of *L. vannamei*. Track 1 (from the outer-ring): 44 linkage groups (LGs) of the shrimp genome. Track 2: Scaffolds anchored to each linkage group. Track 3: Protein-coding genes present in the scaffolds. Red represents genes on forward strand and green for genes on reverse strand. Track 4: Distribution of gene density with sliding windows of 1 Mb. Higher density is shown in darker red color. Track 5: Distribution of GC content in the genome. Track 6: Distribution of SNPs density with sliding windows of 5 Mb. Higher density is shown in darker blue color. Track 7: Distribution of six significantly expanded gene families in the genome, which are opsin (O), peritrophin-like protein (P), chitinase (H), calcified cuticle protein (A), crustacyanin (C), and JHE-like carboxylesterase 1 (J). Track 8: Distribution of SSRs in the genome. Higher density of SSRs is shown with deeper color. Track 9: Distribution of miRNA in the genome. Clusters of co-transcribed miRNAs at adjacent positions are displayed in stacked style. Track 10: Schematic presentation of major interchromosomal relationships in the shrimp genome. **b** SSR content among different animal genomes. **c** Distribution of different types of SSRs in the *L. vannamei* genome. **d** Fluorescence in situ hybridization (FISH) of $(AACCT)_n$ type SSR to the nucleus (i) and the chromosomes (ii) of *L. vannamei*. SNP single nucleotide polymorphism, SSR simple sequence repeat

with $(AT)_n$, $(AC)_n$, and $(AG)_n$ accounting for 81.40% of total SSRs (Fig. 1c, Supplementary Fig. 5). Most SSRs were located in intergenic regions (24.63%) and introns of protein-coding genes (22.07%), and far fewer were found in exons (1.41%). $(AACCT)_n$, a telomere component identified by fluorescence in situ hybridization (Fig. 1c), is longer than many other SSRs, with the longest SSR (13,769 bp), belonging to this type (Supplementary Fig. 6). Furthermore, the length of $(AACCT)_n$ located in introns is longer than those found in other genomic regions (Supplementary Fig. 7).

Transposable elements (TEs) account for 16.17% of the *L. vannamei* genome, with DNA transposons (9.33%) and long interspersed elements (LINEs, 2.82%) comprising the two major classes. En-Spm (6.39%) was found to be the most abundant TE, with its abundance markedly higher than in *D. pulex* (0.05%), *P. virginalis* (0.01%) and *P. hawaiensis* (0.25%)[10–12] (Supplementary Table 12). Most TEs in *L. vannamei* showed higher divergence (substitution rate 19−33%) than those in other crustaceans (Supplementary Fig. 8). However, LINEs in *L. vannamei*, especially RTE-BovB and Penelope, showed a low divergence (Supplementary Fig. 8).

In total, 25,596 protein-coding gene models were annotated in *L. vannamei*. Compared to *D. pulex* and *P. hawaiensis*, *L.*

*vannamei* has a longer average exon size (259 bp) and more exons per gene (5.94) (Supplementary Fig. 9, Supplementary Table 13). We also annotated noncoding small RNAs, including 1458 tRNAs, 464 rRNAs, 296 small nuclear RNAs (snRNAs), 255 small nucleolar RNAs (snoRNAs), 90 ribozymes, and 214 microRNAs (miRNAs) (Supplementary Table 14).

**Genome evolution**. Phylogenetic analysis indicates that crustaceans and hexapods form a monophyletic group of Pancrustacea, with the latter nested within the former, making Crustacea paraphyletic (Fig. 2a). As members of Malacostraca, *L. vannamei* (Decapoda) and *P. hawaiensis* (Amphipoda) diverged at about 240 MYA (million years ago), i.e., in early Mesozoic. This divergence occurred after the Permian-Triassic mass extinction, when about 96% of marine species became extinct[14], followed by the radiation of many new life forms. This divergence period is also consistent with the radiation of shrimp-like decapods in the early Middle Triassic[15]. We found three prominent genome characteristics from *L. vannamei* that might underlie the rapid evolution of penaeid shrimp, namely, abundant SSRs, a high proportion of taxon-specific genes, and extensive tandem gene duplications.

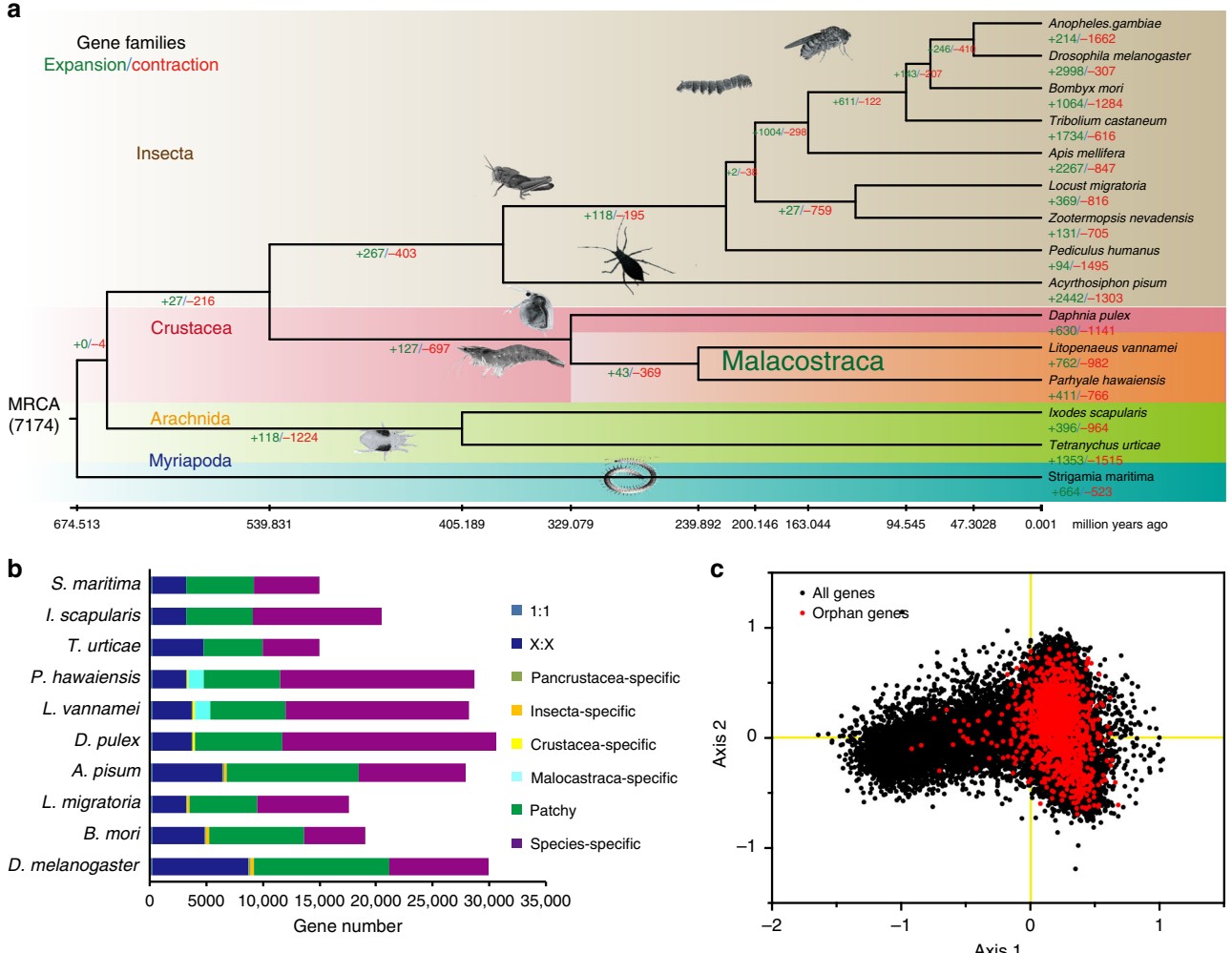

**Fig. 2** Comparative genomics analysis of *L. vannamei* and other arthropods. **a** Phylogenetic placement of *L. vannamei* in the arthropod phylogenetic tree. Numbers on branches indicate the number of gene gains (+) or losses (−). The estimated divergence times are displayed below the phylogenetic tree. Image credits: Gewin V, Martin Cooper, Bernard Dupont, Carlo Brena, Gilles San Martin, David Ludwig, S. Rae. **b** Comparison of the gene repertoire of ten arthropod genomes. "1:1" indicates single-copy genes, "X:X" indicates orthologous genes present in multiple copies in all the ten species, where X means one or more orthologs per species, "patchy" indicates the existence of other orthologs that are presented in at least one genome. **c** Principal component analysis (PCA) on relative codon usage of orphan genes

Because of their special mutational and functional qualities, SSRs play a major role in generating genetic variation underlying adaptive evolution[16]. Besides *L. vannamei*, two penaeid species, *Penaeus monodon* and *Marsupenaeus japonicus*, also have a high proportion of SSRs (~10%) in their genome, traceable through low coverage genome sequencing[17]. However, the genomes of the marbled crayfish *P. virginalis* and the amphipod *P. hawaiensis* contain only 0.99 and 1.27% of SSRs, respectively (Supplementary Table 12). Therefore, the expansion of SSRs might stem from the common ancestor of the penaeid shrimp. SSRs play critical roles in regulating genome plasticity (including DNA recombination and replication) and gene expression[18]. SSRs in *L. vannamei* were widely distributed among introns (22.07%) in 16,741 genes (65.47%), whose expression may be regulated by SSR polymorphism. SSRs may also contribute to DNA recombination with TEs, as most TEs and their up- or downstream sequences contain SSRs (Supplementary Fig. 10), which make up more than 90% for ERV1, Charlie, Sola, MuDR, and En-Spm. Thus, the significant expansion of SSRs might provide a unique genetic architecture for the shrimp's adaptive evolution.

Compared with other crustaceans, the *L. vannamei* genome has 762 expanded gene families and 16,291 species-specific genes (>57% of the entire gene repertoire, Fig. 2b), including genes related to myosin complex, chitin binding, metabolic process, and signaling transduction (Supplementary Data 1–2, Supplementary Tables 15–16, Supplementary Fig. 11). Orphan genes can contribute to lineage-specific adaptation[19]. A total of 3369 orphan genes were identified in the *L. vannamei* genome. They displayed a lower number of exons (4.47 exons/gene), but a greater length of exons (292 bp/gene) in contrast to the average length of all genes (5.94 exons and 260 bp per gene). They exhibited special gene structure characters, and special temporal/spatial expression patterns, suggesting their independent de novo origins (Fig. 2c, Supplementary Figs. 12 and 13). When searched against the transcriptome unigenes of other decapods, the orphan genes were found to be present in other penaeids, such as *Fenneropenaeus chinensis* (83.07%) and *P. monodon* (64.59%), but rarely present in other decapods, e.g. *Exopalamon carincauda* (1.95%), *Eriocheir sinensis* (1.09%), and *Neocaridina denticulata* (3.82%), suggesting that these orphan genes are lineage-specific, and may contribute to penaeid-specific adaptation.

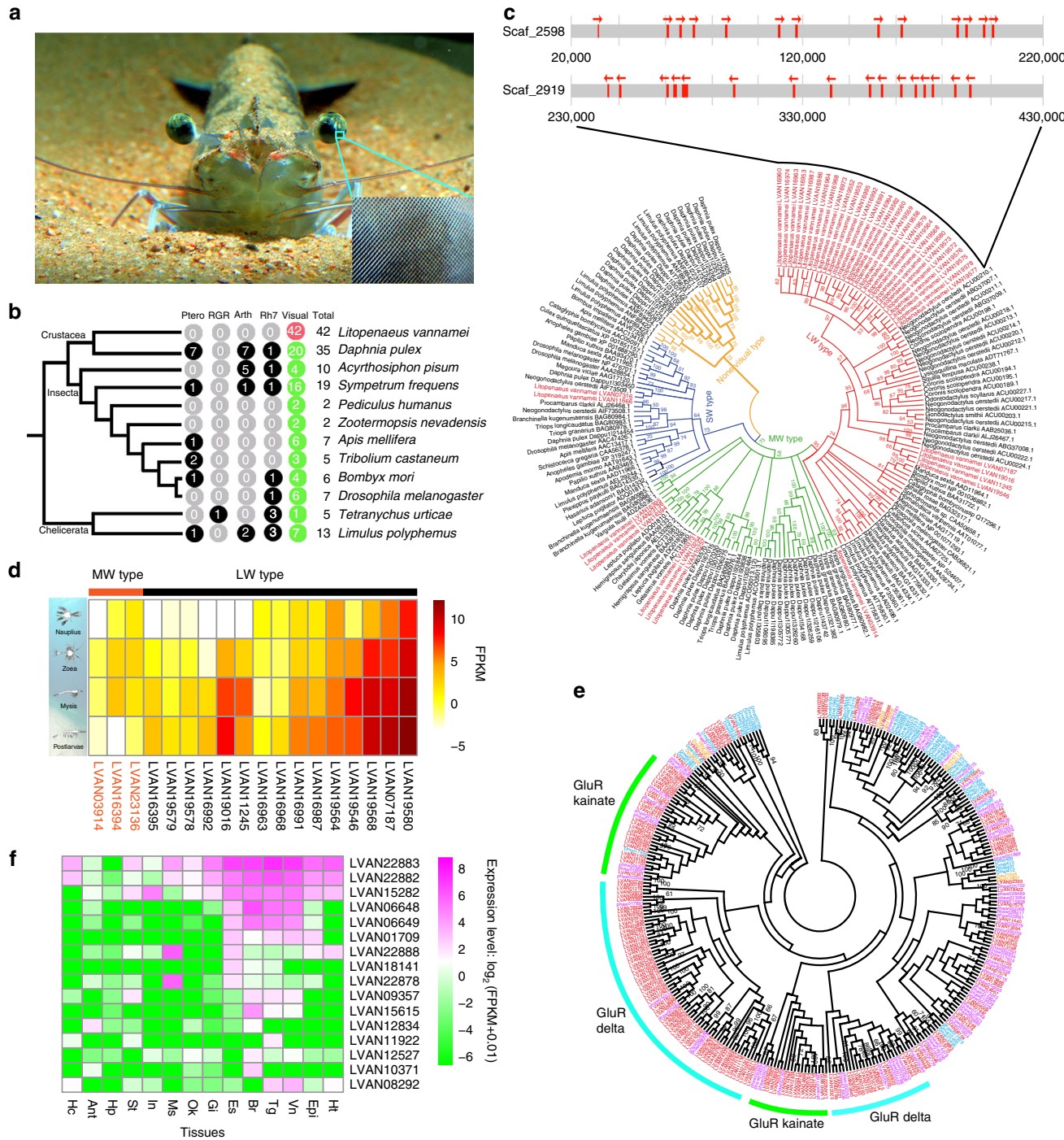

**Fig. 3** Opsin and iGluR gene family in the shrimp genome. **a** Picture of the *L. vannamei* eyes and ommatidia. **b** Opsin genes of *L. vannamei* in comparison with those in the genomes of various arthropods. The number of visual and nonvisual type opsin genes, including pteropsin (Ptero), RGR-like (RGR), arthropsin (Arth), Rh7-like (Rh7) were identified using sequence alignment with known opsins and GPCR-domain searches (Supplementary Fig. 18). **c** Phylogenetic tree of the opsin gene family in arthropods. Six clades of opsins in the *L. vannamei* genome were observed (red). The genes in largest clade are specifically expanded opsins in the *L. vannamei* genome, which are also tandemly duplicated. The arrow indicates the transcriptional orientation. **d** Expression of opsin genes (FPKM>1) during different larval stages of *L. vannamei*. **e** Expansion of ionotropic glutamate receptor (iGluR) genes in *L. vannamei*. **f** Expression of iGluR genes in different tissues: hemocyte (Hc), antennal gland (Ant), hepatopancreas (Hp), stomach (St), intestine (In), muscle (Ms), lymphoid organ (Ok), gill (Gi), eyestalk (Es), brain (Br), thoracic ganglion (Tg), ventral ganglion (Vn), epidermis (Epi), heart (Ht). GPCR G protein-coupled receptor

Gene tandem duplication was one of the most intriguing features of the *L. vannamei* genome. A total of 4662 genes were identified to be tandemly duplicated (Supplementary Fig. 14), and these duplicates share high sequence similarities (identity >98%), indicating recent duplications. Moreover, these genes mainly consist of opsins, crustacyanins, chitinases, cuticle proteins, myosins, actins and heat shock proteins, which may be important for adaptive evolution of the shrimp.

**Visual system for benthic adaptation**. *L. vannamei* has a pair of large, stalked eyes with an ovoid visual surface making up of a

highly elevated number of ommatidial lenses[1] (Fig. 3a). Each compound eye consists of about 55,000–80,000 ommatidia[20], rivaling the most acute eyes in arthropods. Adult and juvenile *L. vannamei* are benthos in turbid and shallow waters to a depth of 50 m, where no light with a wavelength <500 nm penetrates. Therefore, *L. vannamei* are expected to have developed

photoreceptors capable of capturing photons most common in this environment[21]. The genome sequencing of *L. vannamei* now allows us, for the first time, to infer genomic-level evolution in this delicate eye[1]. We identified 42 opsin genes, which are the key light-sensing factors responsible for visual signal transduction (Fig. 3b). All these opsins belong to the visual type related to color

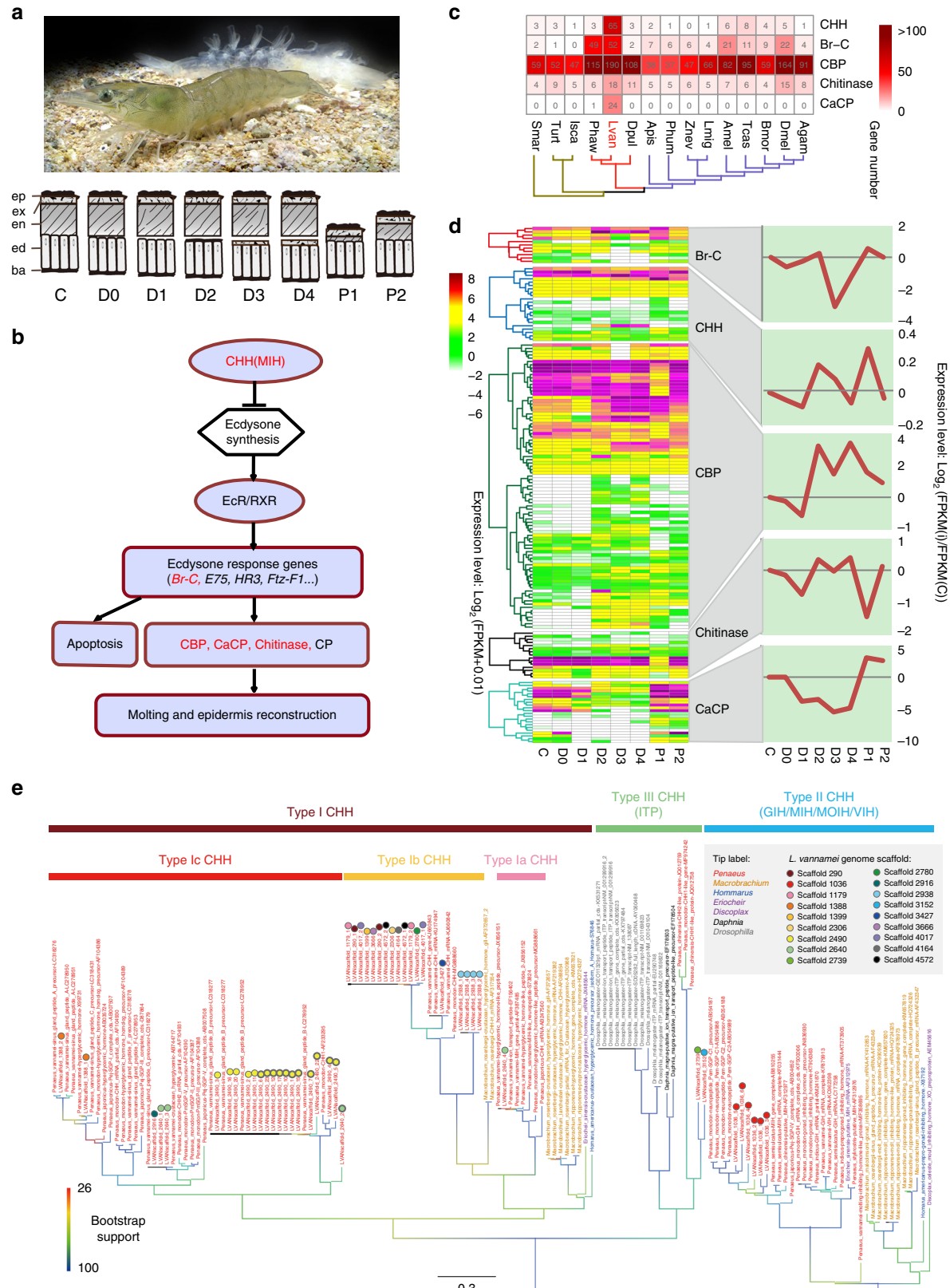

sensation. To our knowledge, *L. vannamei* has the most visual opsin genes among animals, including creatures generally regarded as having excellent vision such as mantis shrimps (6–33 opsins) and dragonflies (20 opsins)[22]. The 42 visual-opsins can be divided into three subfamilies, 33 in the long-wavelength type (LW) subfamily, seven in the middle-wavelength type (MW) subfamily and two in the short-wavelength type (SW) subfamily (Fig. 3c). Intriguingly, most of the LW subfamily genes belong to two clusters in the genome (Fig. 3c), suggesting a major expansion via tandem duplication.

The expansion of LW opsins sensitive to light at the 560 nm wavelength facilitates feeding of the shrimp in benthic environment with predominantly yellow and red (long-wavelength) light[21]. Expansion of opsin genes was also observed in the genome of the water flea *D. pulex* (36 opsins), but this expansion is restricted to the MW subfamily, which is sensitive to light at ~530 nm wavelength associated with planktonic habitat (Supplementary Fig. 15). Considering the shift from planktonic to benthic habitat during the shrimp's life cycle, we analyzed the expression of opsin genes at different larval stages of *L. vannamei*. Interestingly, LW-type opsin genes showed significantly higher expression levels in the benthic postlarvae, whereas the MW opsin genes displayed higher expression in the planktonic larvae (Fig. 3d). These data provide strong support for the correlation of differential expression among the types of visual opsin genes with the planktonic-benthic habitats, suggesting that *L. vannamei* has a highly effective visual system adapted to its habitat transition.

**Nerve signal conduction and locomotion**. In the central nervous system of decapods, the giant nerve fibers originating from the brain and terminating at the caudal ganglion are responsible for the rapid escape reactions[23]. These nerve fibers of penaeid shrimp have a conduction speed of ~200 ms⁻¹, generally regarded as the fastest conducting speed in animals[24], compared with a mere 20 ms⁻¹ for squids, which have the largest nervous system among invertebrates[25,26]. In the *L. vannamei* genome, the gene categories for signal recognition and neural development are significantly enriched (Supplementary Data 1, Supplementary Tables 16). Marked expansion was found in several gene families, including 457 G protein-coupled receptors (GPCRs), 53 transient receptor potential channels, 21 innexins, and 47 protocadherins (Supplementary Table 17), which are considered as the core neural-development-related genes across bilaterians[26]. The rapid nerve conduction in shrimps is based on their heavy myelin sheath[27], but the myelin substrate, cholesterol, cannot be synthesized in most invertebrates[25,28]. Crustacyanins, the apolipoprotein family genes which transport cholesterol, are significantly expanded and tandemly duplicated in the *L. vannamei* genome, and most of them are highly expressed in the neural and digestive system (Supplementary Data 3).

Unlike vertebrates and most protostomes that use acetylcholine as a neurotransmitter, arthropods use glutamate at the neuromuscular junction[29]. A total of 169 ionotropic glutamate receptors (iGluRs) and 148 glycine receptors (GlyRs) genes are present in the *L. vannamei* genome (Fig. 3e, Supplementary Fig. 16, Supplementary Table 18). iGluRs are involved in excitatory neurotransmission, which can occur quickly and last for a long time. They also play a role in regulating myelination[30]. GlyRs can activate Cl⁻ conductance in neurons[31]. Furthermore, iGluRs and GlyRs genes are mainly expressed in the nervous system as well as in muscle (Fig. 3f, Supplementary Fig. 17). In addition, some gene families involved in muscle contraction, such as actins and myosins, are also significantly expanded in the *L. vannamei* genome (Supplementary Table 16). These features might equip the shrimp to rapidly perform neural signal transduction, enhancing its responsiveness and strengthening its locomotion, including escape reactions in adaptation to its benthic swimming life in shallow seawater.

**Intensified ecdysone signal pathway**. Molting is a basic physiological process of crustaceans (Fig. 4a), through which they realize metamorphosis, growth, and development. An ecdysone signal pathway, under hormonal regulation, controls every step in molting (Fig. 4b). Both an initial rise and a coordinated decline of circulating ecdysone concentration are necessary for successful molting[32]. In *L. vannamei*, the immediate function of ecdysone in promoting molting is amplified by expanded downstream genes in the ecdysone signal pathway. After entering the nucleus, ecdysone activates successive transcription of early stage ecdysone response genes (Fig. 4b). The major ecdysone response gene, *Broad-Complex* (*Br-C*), was perceptibly expanded in the genome (Fig. 4c), and were highly expressed at early premolt (D1) and postmolt (P1) (Fig. 4d), consistent with their function in regulating the expression of downstream effectors, such as chitinase and cuticular proteins[33]. Cuticle proteins, chitin binding proteins (CBPs) and calcification-related cuticular proteins (CaCPs) are main structural proteins for constructing the cuticle[34], which protects the shrimp from bacteria, fungi, viruses, and mechanical injuries. However, to allow for growth in body size, the cuticle is degraded prior to ecdysis by chitinase[34]. Genes encoding CBPs, CaCPs, and chitinase are significantly expanded in the *L. vannamei* genome (Fig. 4c, Supplementary Table 19). These genes were mainly expressed in the epidermis and intestinal peritrophic matrices (Supplementary Fig. 19). The structural proteins were coordinately expressed during molting, which began with the initial degradation by chitinase, and then the synthesis of new cuticle with high expression of Rebers and Riddiford 2 (R&R2)-type cuticle protein and CBPs genes from early premolt (D2) to late premolt (D4). After ecdysis (P1–P2), the CaCPs and R&R1-type cuticle proteins were produced to harden the cuticle (Fig. 4d,

**Fig. 4** Key gene families related to the ecdysone signal pathway in the *L. vannamei* genome. **a** Picture of shrimp and its shed exoskeleton. Schematic diagrams below the picture show the changes of the epidermis (ep: epicuticle, ex: exocuticle, en: endocuticle, ed: epidermis, ba: basement membrane) during the molting cycle. The molting cycle can be divided into eight stages: Intermolt (C) stage, Premolt stages (D0–D4), and Postmolt stages (P1–P2). **b** The shrimp ecdysone signal pathway. Expanded gene families are highlighted in red. **c** Distribution pattern of the expanded gene families in shrimp ecdysone signal pathway and comparison with other arthropods. **d** Expression patterns of the expanded gene family genes at the different molting stages of *L. vannamei*. The overall expression trends in different molting stages are shown in the right. **e** Phylogenetic analysis of the CHH family based on CDS sequences containing full CHH domain, reconstructed using IQ-tree 1.6.2 under TVMe + I + G4 model with ultrafast bootstrap method. Sequences from *L. vannamei* genome are annotated with circles of various colors to indicate their corresponding scaffold of origin. GenBank sequences of the CHH family from genus *Penaeus* s.l., *Macrobrachium*, *Homarus*, as well as from species *Eriocheir sinensis*, *Discoplax celeste*, *Daphnia magna* and *Drosophila melanogaster* were incorporated in this analysis. These sequences were annotated in the phylogenetic tree as follows: species name/gene description/accession number. They were classified into type I, II, and III CHH subfamilies, in which type III CHH consisted of ion transport peptides (ITPs). CHH crustacean hyperglycemic hormone

Supplementary Fig. 19). The expanded genes encoding Br-C and cuticular proteins in the genome can enhance the functioning of ecdysone, once its concentration reaches a certain threshold. The enhanced ecdysone function might be the primary mechanism to assure the frequent molting behavior of shrimps.

In *L. vannamei*, a striking expansion of the crustacean hyperglycemic hormone (CHH) family underpins the control of the enhanced ecdysone function (Fig. 4c). Ecdysone synthesis in the crustacean Y-organ is mainly under the negative control of the molt-inhibiting hormone (MIH), one of the type II peptides of the CHH family[35]. We identified seven tandemly duplicated MIH genes (Fig. 4e) with high sequence similarity (Supplementary Fig. 20). This might facilitate the coexpression and cofunctionality of these genes during molting. MIH transcripts were mainly located in the eyestalk and ventral nerve (Supplementary Data 4), and their expression was low at late premolt and high from postmolt to early premolt (Supplementary Data 4), the exact reverse of the hemolymph ecdysone level[36]. Besides, type I peptides of the CHH family are also known to suppress ecdysteroidogenesis in the Y-organ, though at a higher dose than MIH[37]. In the *L. vannamei* genome, we identified three major clades of type I CHH peptides: (1) type Ia CHH clade, which contains peptides more highly expressed in gut, thoracic ganglion and lymphoid organ than in eyestalk and other tissues (Supplementary Data 4), and may function in ion/osmoregulation and water uptake but lack hyperglycemic functions, (2) type Ib clade, which is highly expressed in eyestalk as well as brain and thoracic ganglion, and is less well-studied but also believed to assume an ion/osmoregulatory function[38], and (3) type Ic clade, which is penaeid-specific and contains peptides involved in the regulation of molting, reproduction, energetics and ionic metabolism[39] (Fig. 4e). Significantly, we detected much more striking expansion in the type Ic CHH than in other type I CHH genes in the genome (Fig. 4e, Supplementary Data 5). The type Ic CHH genes showed similar expression patterns as MIH genes in different tissues and molting stages, while the other type I CHH did not (Fig. 4d, Supplementary Data 4). Most type Ic CHH genes are tandemly located in Scaffold 2490 (Fig. 4e) and share high sequence similarity (Supplementary Fig. 21). Similar gene organization and expression patterns between the type Ic CHH genes and MIH genes suggest that they might play similar roles in molting. Furthermore, MIH and CHH also increase the intracellular concentration of cyclic nucleotides (cAMP or cGMP) and $Ca^{2+}$ in the Y-organ, which further inhibits ecdysteroidogenesis[40] by activating the "triggering" phase of MIH regulation[50]. The genes involved in the "triggering" phase of MIH regulation, including cAMP-dependent protein kinase, phosphodiesterase 1, calmodulin and $Ca^{2+}$/calmodulin-dependent protein kinase genes, were all under positive selection (Supplementary Fig. 22, Supplementary Data 6). The expansion and tandem duplication of CHH family genes, and positive selection of MIH "triggering" phase genes might strengthen the negative control of ecdysone synthesis. Collectively, this intensified ecdysone signal pathway, generated by gene expansion, tandem duplication and positive selection, with amplification of ecdysone function and negative control on ecdysteroidogenesis, serves to ensure the frequent and precise molting process in the shrimp.

**Key genes mediating environmental factors in molting regulation**. Molting is influenced by many environmental factors, such as nutrition and photoperiod. MIH genes of *L. vannamei* universally contain the sterol regulatory elements (SRE), i.e. the binding sites for SRE-binding protein (SREBP), in the regulatory sequence (Supplementary Fig. 23). Furthermore, all detected transcripts of *MIHs* and *SREBP* had similar expression patterns,

with high expression levels in the eyestalk at intermolt and low levels at late premolt (Fig. 5a). SREBP is a major regulator of cholesterol metabolism, and its function is inhibited by a high level of cellular cholesterol[41]. Ecdysone is biosynthesized with cholesterol as the substrate, and cholesterol cannot be synthesized in vivo in crustaceans[28]. The ecdysone level[46] exhibits a similar trend to the in vivo cholesterol level during molting[53], and both show opposite trends to expression patterns of *MIHs* and *SREBP*. These observations suggest that a high cholesterol level may drive molting through SREBP, which may positively regulate *MIH* expression, thus suppressing ecdysone synthesis and molting. This hypothesis was supported by a reduction (~95%) in expression of all detected MIH genes after *SREBP* silencing in the eyestalk (Fig. 5b, Supplementary Fig. 24). Injection of cholesterol into shrimps had a similar inhibitory function on *MIH* expression, reducing *MIH* expression levels by >90% (Fig. 5b). This inhibition was possibly exerted through inhibiting nuclear translocation of *SREBP*[42]. *MIH* silencing (Supplementary Fig. 24) significantly accelerated molting of shrimps, making them develop to the late premolt (D3, Fig. 5c, i). However, most control shrimps at intermolt progressed to early premolt (D1) within the same duration (Fig. 5c, ii). By contrast, injection of Lipitor, a lipid-lowering agent, apparently retarded molting at the D3 stage (Fig. 5c, iii), as control shrimps progressed to postmolt (Fig. 5c, iv). These data indicated that *SREBP* silencing and a high cholesterol level had similar functions as *MIH* silencing, supporting the hypothesis that SREBP mediates MIH-regulated ecdysteroidogenesis.

The photoperiod has evident effects on crustacean molting[43]. Due to the key function of opsin in visual transduction, we speculate that opsin might be the primary mediator between photoperiod and molting. This is supported by the upregulated levels of MIH genes after silencing of several LW opsin genes (Fig. 5b, Supplementary Fig. 25), indicating an inhibitory role of opsin in *MIH* expression. In addition, after 10 days of opsin silencing, shrimps exhibited a lower molting rate than the control (Fig. 5d).

In sum, we infer that cholesterol supply and photoperiod regulate molting via SREBP and opsin, respectively (Fig. 5e). In this model, a low cholesterol level corresponds to increased *SREBP* and *MIH* expression at postmolt. Cholesterol uptake leads to inactivation of SREBP and consequent reduction in *MIH* expression at early premolt. Simultaneously, ecdysone also rises to initiate the downstream signaling pathways. The model also proposes the function of opsin in mediating the effect of photoperiod through modulating the MIH-regulated molting.

**Immune protection during molting**. The newly formed cuticle of shrimps at postmolt is thin and not yet hardened, making them prone to pathogen infection. We anticipate that the shrimp may possess an intensified immune protection during ecdysis. Crustaceans mainly rely on their innate immune system, consisting of humoral immunity and cellular immunity, to defend against pathogens[44]. We found 111 immune-related genes with differential expression patterns at different molting stages. These genes were clustered into six groups according to their functions, including 15 antimicrobial peptides (AMPs), 59 pattern recognition receptors (PRRs), six phenoloxidase-system-related genes, 14 apoptosis-related genes and 17 homeostasis-related genes (Supplementary Data 7).

Crustins contributed to 13 of the 15 differentially expressed AMPs (Supplementary Data 7). Most of them showed the lowest expression levels during late premolt and increased expression levels immediately after ecdysis (Supplementary Fig. 26A). Crustins were mainly distributed in the epidermis,

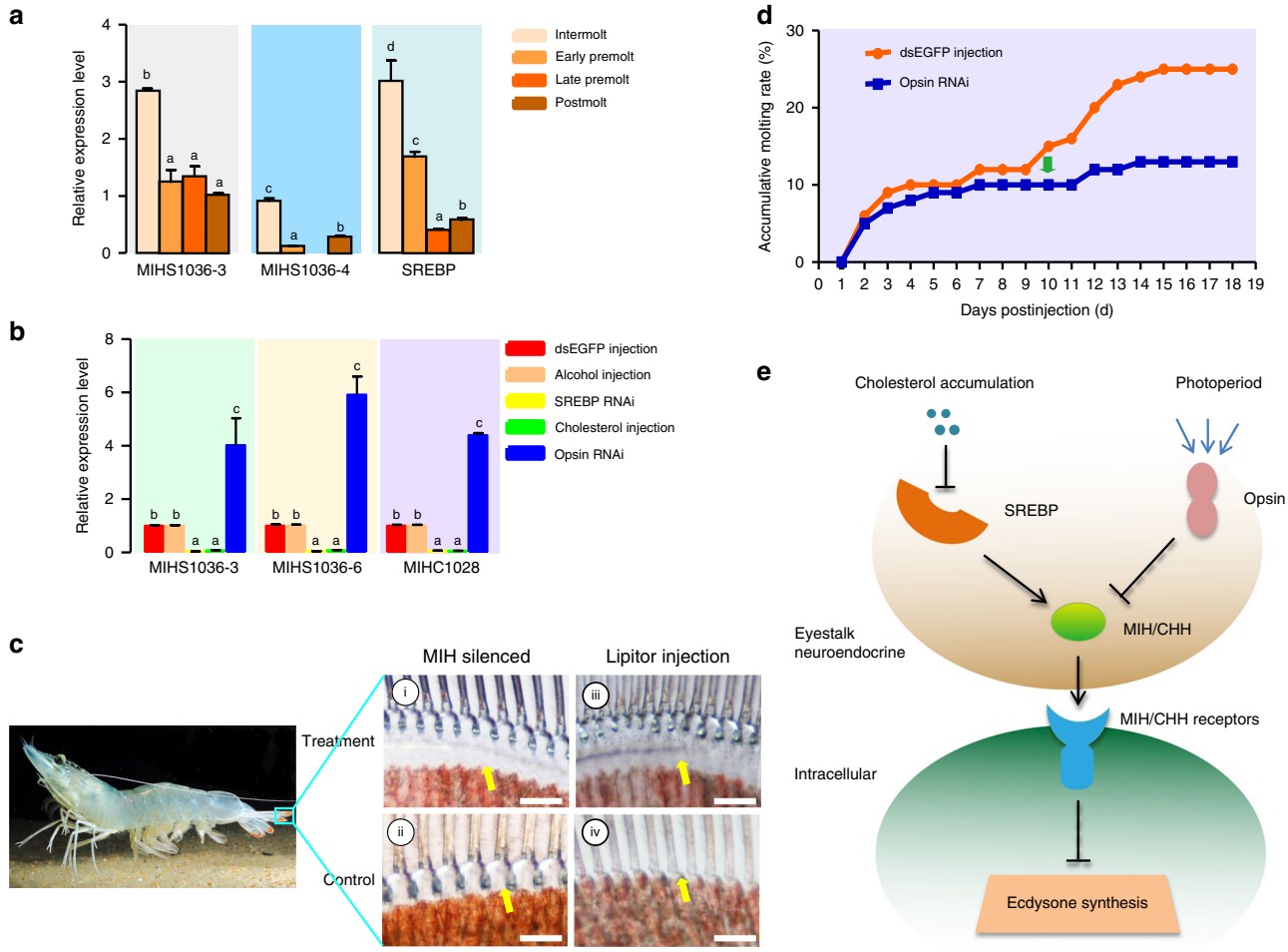

**Fig. 5** Regulation of MIH-mediated ecdysone synthesis in shrimp. **a** The expression patterns of *SREBP* and *MIH*s in shrimp eyestalks at different molting stages. **b** The expression profiles of *MIH*s in the eyestalk with different treatments. Injections of dsEGFP and PBS serve as controls. **c** Phenotype changes of the shrimp uropods cuticle after treatments. Shrimps at the intermolt stage were treated by RNAi: (i) 2 days after injection of dsMIH; (ii) 2 days after injection of dsEGFP. Shrimps at the early premolt stage were used for Lipitor treatment: (iii) 2 days after injection of Lipitor; (iv) 2 days after injection of PBS. Yellow arrows indicate the space between old and new cuticles at different molting stages. Bar = 50 μm. **d** Effect of opsin silencing on accumulative molting rates, with the injection of dsEGFP serving as a control. The green arrow shows the time when the treatment led to a significantly lower accumulated ecdysis rate compared to the control group. **e** The schematic diagram of the regulation of SREBP and opsin on the MIH-mediated ecdysone synthesis. SREBP SRE-binding protein

stomach, gill, and hemocytes (Supplementary Fig. 26B). Of the 59 identified PRRs, 53 are lectins, important PRRs involved in the innate immunity of crustaceans[45] (Supplementary Data 7). Several lectin genes were highly expressed at P1 stage (Supplementary Fig. 26C), mainly detected in the epidermis, stomach and gill (Supplementary Fig. 26D, 27). Most other lectins were highly expressed in the hepatopancreas, which exhibited the lowest expression levels at the P1 stage and was highly expressed from P2 to early premolt (Supplementary Fig. 28). Expressions of the phenoloxidase-system-related genes apparently increased at postmolt (Supplementary Data 7) and were mainly distributed in the gill, epidermis, stomach, intestine, hemocytes, and hepatopancreas (Supplementary Fig. 26E, F, G). Recently, a phenoloxidase gene in insects was reported to play a role in skin immunity, protecting the host against fungal infection after ecdysis[46]. Most of the identified apoptosis-related genes, mainly expressed in the intestine and hepatopancreas, were highly expressed throughout the molting process, with two peaks at D2 and D4 (Supplementary Fig. 26E, F). At premolt, a new cuticle is generated after apolysis, when the old cuticle is separated from the epidermis. The expression patterns of apoptosis-related genes indicated that they might be involved in apolysis rather than immunity.

In summary, AMPs (crustins, anti-lipopolysaccharide factor and penaeidin), PRRs (lectins and LGBP) and phenoloxidase system genes largely contribute to the immune protection during ecdysis (Supplementary Fig. 26H). While these immune-related genes are important in protecting the host against pathogens in the external environment, their high expression in the hepatopancreas might be related to the in vivo gut flora stability of the shrimps and protect the host against pathogens in internal environment as bacteria, including some pathogens are ubiquitous in the hepatopancreas.

**Impacts of breeding**. *L. vannamei* became an important aquaculture species in the 1980s, and a selective breeding program began under the "US Marine Shrimp Farming Program (USMSFP)" in 1989[47]. Compared to the long history of domestication in plants, terrestrial livestock and poultry[48,49], the history of *L. vannamei* breeding is relatively short. However, the selection pressure in *L. vannamei* is very high, as the shrimp could be

selected one generation per year and a single spawning female can produce >300,000 offspring[50]. To examine the genomic consequences of such intensive selective breeding, we selected 22 individuals for genome resequencing, including 8 individuals from the wild and 14 individuals from five different broodstocks (Supplementary Table 20). We sequenced them with an average of >23× genome coverage (Supplementary Table 21), and identified 31,993,474 single nucleotide polymorphisms (SNPs) among the shrimps (Supplementary Table 22), with an average of 19 SNPs/Kb. The SNPs density in the *L. vannamei* genome is much higher than that in chickens[51] and pigs[52], and similar to that in the oyster, which has extremely high level of heterozygosity[6]. Most SNPs are located at the intergeneic regions (86.68%), and the coding regions display much lower genetic diversity than the introns. In the coding regions, 261,056 synonymous and 206,026 nonsynonymous SNPs were identified (Supplementary Table 23). Around 1400 SNPs are located at splice acceptor sites and 1452 SNPs at splicing donor sites. To our knowledge, this represents the largest set of high-quality SNPs obtained from *L. vannamei*, and it would constitute a valuable resource for genetic research and selection.

Significant differences were observed between the genetic diversity of wild shrimp and various broodstocks. The average nucleotide diversity ($\pi$) at the whole-genome level for the two groups was $3.69 \times 10^{-3}$ and $2.70 \times 10^{-3}$, respectively. Phylogenetic analyses based on the whole-genome SNPs separated the wild individuals from those in the broodstocks, which were then grouped into two clades, with the individuals in Ecuador clustered separately from the other commercial breeding lines, apparently because the former only underwent four generations of selection (Fig. 6a).

Comparison of the divergence index ($F_{ST}$) and the $\theta\pi$ ratios ($\pi_{wild}/\pi_{breeding}$) between the wild and cultured populations allowed us to identify 14 regions (1.66 Mb in size) (Fig. 6b, c), containing 28 predicted genes (Supplementary Table 24), that are under strong selection. GO and KEGG enrichment analyses show that Rho protein signal transduction pathway and glycosphingolipid biosynthesis pathway are enriched. Two copies of *A-kinase anchor protein 13* (*AKAP13*), involved in the Rho protein signal transduction pathway, are strongly selected (Fig. 6c, Supplementary Fig. 29). AKAP13 is a scaffold protein with guanine exchange factor activity and plays a role in TLR2-mediated NF-κB activation, which is involved in innate immunity[53]. The selected gene in the glycosphingolipid biosynthesis pathway is annotated as lactosylceramide 4-alpha-galactosyltransferase (A4GALT) (Fig. 6c). *A4GALT* is involved in protein glycosylation and protein modification process necessary for synthesis of the receptor for bacterial verotoxins in mammals[54]. Further, one type of anti-lipopolysaccharide factor like protein (ALF) (Supplementary Fig. 29) that is essential to immune defense[55] was also found to be under strong selection. These data illustrate that artificial selective breeding under culture conditions influences the disease resistance of shrimp.

## Discussion

The high-quality genome assembly of *L. vannamei* provides the opportunity to understand various biological processes of shrimp at the genome level. The most prominent characteristic of the coding region of the shrimp genome is the expansion of a series of genes related to visual system, nerve signal conduction, and molting. These gene expansions might have provided the shrimp with its excellent eyesight and rapid nerve signal conduction, better equipping it to adapt to its benthic habitat. Our genome assembly also sheds light on the molting process and provides important evidences for exploring the similarities and differences

between crustaceans and other ecdysozoans, including insects. As a vital process in shrimp, molting is tightly linked to growth, metamorphosis and reproduction. Elucidating the regulatory mechanisms of molting will provide useful clues for genetic analysis of these important processes.

Our genomic analyses have also revealed that almost 30 years of breeding has exerted a significant impact on the genome of the *L. vannamei* broodstocks. An increase in aquaculture production and efficiency can be achieved through genetic improvement of cultured stocks. The assembled shrimp genome and the large amount of SNP markers will provide a useful resource for the application of genome-wide association studies and genomic selection, and thus accelerate genetic improvements in shrimp culture.

## Methods

**Genome sequencing**. The muscle of a single male adult of *L. vannamei* (Kehai No.1 variety, Hainan, China) was used for DNA extraction and genome sequencing. Genomic DNA was prepared using TIANamp Marine Animal DNA Kits (TIANGEN, Beijing, China). The DNA was sheared using a sonication device for paired-end (PE) library construction. Libraries with insert sizes ranging from 200 bp to 20 Kb were constructed according to the instructions provided in the Illumina library preparation kit. All libraries were sequenced on HiSeq 2000 and HiSeq 2500 platform (Illumina, San Diego, USA). These raw reads were subsequently trimmed for quality using Trimmomatic v.0.35 (http://www.usadellab.org/cms/index.php?page=trimmomatic). Illumina sequence adaptors were removed, low-quality bases from the starts or ends of raw reads were trimmed, and reads were scanned using a 4-bp sliding-window and trimmed when the average quality per base dropped below 15. The clean reads obtained from this process were used for subsequent analysis.

For PacBio library construction, the genomic DNA of *L. vannamei* was sheared to ~20 Kb and the fragments below 7 Kb were filtered using BluePippin. Filtered DNA fragments were then converted into the proprietary SMRTbell library using the PacBio DNA Template Preparation Kit. In total, ~133 Gb of quality-filtered data were obtained from the PacBio sequencing.

**Genome size estimation**. We adapted a method called Jellyfish[56], which is based on k-mer distribution, to estimate the genome size with the high-quality reads from the short-insert size libraries (500 bp). We obtained a k-mer depth distribution from the Jellyfish analysis and could clearly observe the peak depth from the distribution data. In order to obtain an estimation of the *L. vannamei* genome size, the following formula was applied: genome size = total_kmer_num/kmer_depth, where total_kmer_num is the total number of k-mers from all reads, and kmer_depth is the peak depth.

**Genome assembly**. All of the subreads from the PacBio sequencing were assembled using the WTDBG software (https://github.com/ruanjue/wtdbg-1.2.8). The assembled sequence was then polished using Quiver (SMRT Analysis v2.3.0) with default parameters. To ensure the high accuracy of the genome assembly, several rounds of iterative error correction were performed using the Illumina clean reads.

Finally, the de novo assembly of the PacBio read sequences was carried out using continuous long reads following WTDBG. The paired-end reads of Illumina (200, 300, and 500 bp libraries) were mapped to polish assembly sequence from Quiver by BWA (https://github.com/PacificBiosciences/GenomicConsensus). The SNPs and small insertions and deletions (INDELs) were called and corrected by SAMTools and an in-house script ("SNP_correction.pl", deposited in https://github.com/jianbone/L_vannamei_genome). Finally, we generated scaffolds and performed gap-filling with SSPACE 3.0 with parameter values of "-x 1 -m 50 -o 10 -z 200 -p 1" using meta-paired sequencing data (Illumina 5, 10, and 20 Kb libraries).

**Quality assessment of genome assembly**. To evaluate the quality of the genome assembly, we mapped partial reads from the short-insert size library back to the scaffolds using Bowtie2 with the following parameters: --rdg 3,1 --rfg 3,1 --gbar 2 [57]. A total of 93% of the reads could be mapped back to the current assembly. To evaluate the completeness of the genome assembly in genetic regions, we used conserved core genes by running software CEGMA on the assembly of the *L. vannamei* genome[58].

**Repeat annotation**. Both RepeatModeler and RepeatMasker were used to perform de novo identification and mask of repeats[59]. To ensure the integrity of genes in the subsequent analysis, low complexity or simple repeats were not masked in this analysis, because some of them could be found within the genes.

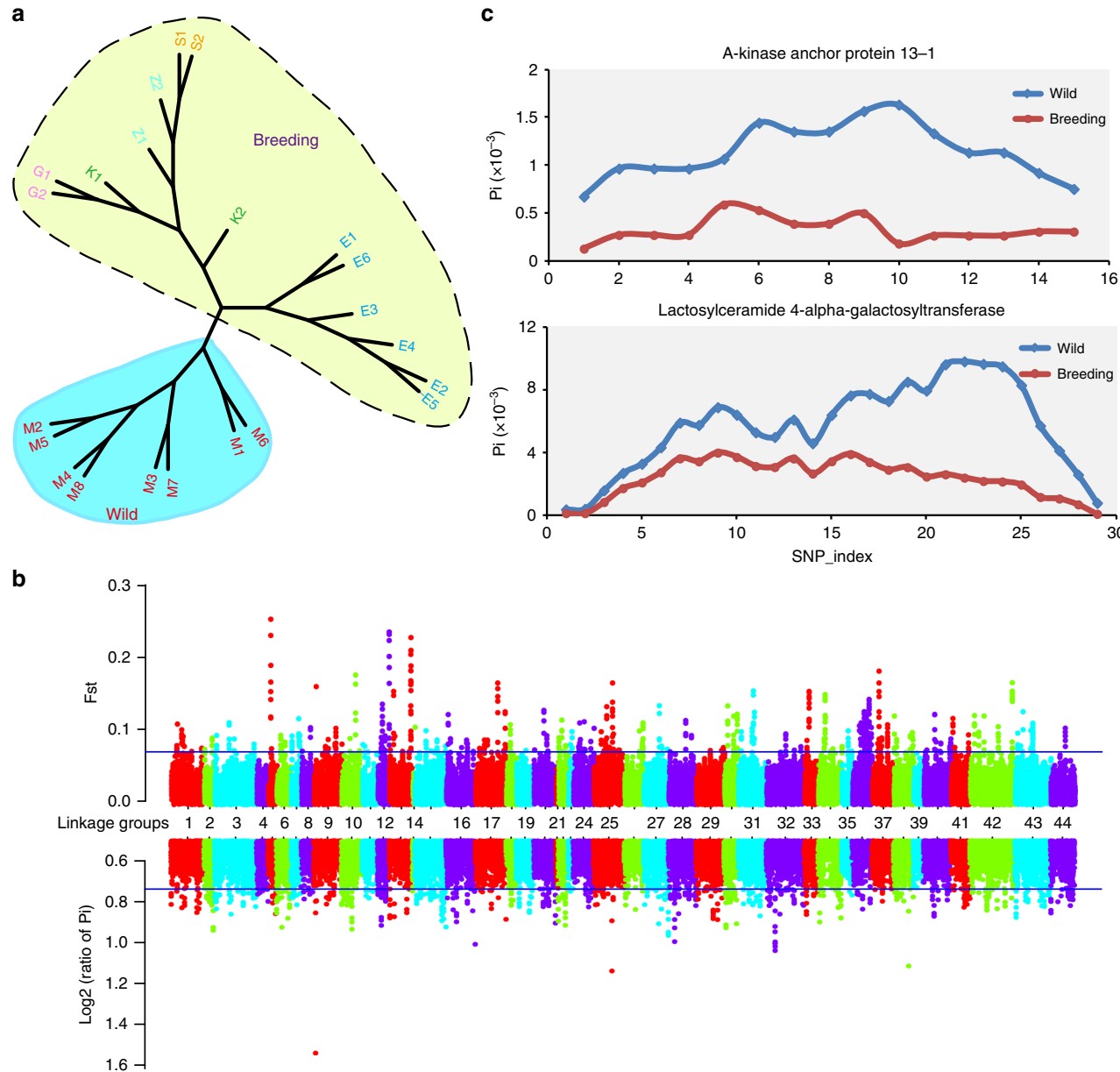

**Fig. 6** Genetic diversity and selective sweep analyses in *L. vannamei*. **a** The phylogenetic tree of wild (blue) and breeding shrimp (yellow) constructed using whole-genome SNPs. M1–M8 represent the wild shrimp collected from Mexico, E1–E6 represent the breeding shrimp collected from Ecuador, K1 and K2 represent the Kehai No.1 variety from China, G1 and G2 represent Guihai No.1 variety in China, S1 and S2 represent Shrimp Improvement broodstock in USA, Z1 and Z2 represent Charoen Pokphand broodstock in Thailand. **b** Genome-wide view of differentiation ($F_{ST}$) and reduction in diversity ($\pi_{wild}$/ $\pi_{breeding}$) statistics associated with the wild and breeding shrimp. The blue line indicates the top 1% of rank level values for empirical percentiles. **c** The genetic diversity value $\theta\pi$ in wild and breeding population for the genes encoding A-kinase anchor protein 13-1 (AKAP13-1) and Lactosylceramide 4-alpha-galactosyltransferase (A4GALT). A sliding-window approach (2 Kb windows with 400 bp increments) was used to calculate $\theta\pi$ of every window for each gene, and the SNP-index indicates the sliding-window of each gene

**Transcriptome sequencing and analyses**. The *L. vannamei* samples at different developmental stages, molting stages, and different adult tissues were collected at the CAS Key Laboratory of Experimental Marine Biology in Qingdao. Total RNA was isolated and purified from all the samples using the TRIzol extraction reagent (Thermo Fisher Scientific, USA), in accordance with the manufacturer's protocol. RNA quality and concentration were assessed using 1% agarose gel electrophoresis, and RNA concentration was measured using Nanodrop 2000 spectrophotometer (Thermo Fisher Scientific, USA). The total RNA of the same tissue was pooled to construct the Illumina sequencing libraries, and then paired-end sequences were generated using Illumina HiSeq 2000 and HiSeq 2500 platform. The clean reads were assembled using Trinity (v2.6.5, https://github.com/trinityrnaseq/ trinityrnaseq/releases) to obtain transcripts with min_kmer_cov set to 2 and all other parameters set to default. The TopHat v1.2.1 package was used to map

transcriptome reads to the *L. vannamei* genome[60]. Cufflinks v2.2.1 (http://cole-trapnell-lab.github.io/cufflinks/) was used to calculate expected FPKM (Fragments Per Kilobase of transcript per Million fragments mapped) as expression values for each transcript.

**Gene prediction and annotation**. Protein-coding region identification and gene prediction were performed through a combination of homology-based prediction, de novo prediction and transcriptome-based prediction methods. Proteins from several species, including *D. pulex*, *P. hawaiensis*, *Drosophila melanogaster*, *Anopheles gambiae*, *Locust migratoria* and *Bombyx mori*, were downloaded from NCBI. Protein sequences were mapped against *L. vannamei* with Exonerate version 2.2.0 (http://www.ebi.ac.uk/~guy/exonerate/). The blast hits were used in predicting the

gene structure of the corresponding genomic regions. The de novo prediction software, Augustus v2.5.5, was used to predict the coding regions in the repeat-masked genome. RNA-Seq data were mapped against the assembly using Tophat v2.1.1[60]. Cufflinks v2.2.1 was then used to convert the transcripts from the result of Tophat to gene models. All gene models derived from these three methods were integrated by EvidenceModeler (EVM) into a nonredundant gene set[61].

Functional annotations of the obtained gene set were conducted using BLASTP with an E-value of 1e−5 against the NCBI-NR, SwissProt databases, and KEGG database. Protein domains were annotated by mapping to the InterPro and Pfam databases using InterProScan and HMMER[62,63]. The pathways in which the genes might be involved were derived from gene mapping against the KEGG databases. The Gene Ontology (GO) terms were extracted from the corresponding InterProscan or Pfam results.

**Noncoding RNA prediction.** Noncoding RNA genes were predicted for the repeat-masked genome by sequence- and structure-based alignments with the Rfam noncoding RNA database (http://xfam.org/), with an E-value cutoff of $1e^{-02}$ using Infernal (http://eddylab.org/infernal/). Specifically, expression criteria were applied for miRNA identification, and small RNA sequencing data of *L. vannamei* were downloaded from the SRA database of NCBI (SRX2648655, SRX2648646, and SRX682234). Adaptor and primer sequences were trimmed, and low-quality sequences were removed. Clean sRNA reads were compared with the Rfam database (http://xfam.org/) to exclude noncoding RNAs other than miRNAs. The remaining sRNA reads were subjected to miRNA identification by mapping to predicted pre-miRNA structures in the *L. vannamei* genome using miRDeep2 and miReap (http://sourceforge.net/projects/mireap/). Conserved miRNAs were identified by BLAST alignment to mature miRNA sequences deposited in the miRBase (http://mirbase.org/), requiring alignment ≥15 nt which covers the miRNA seed region (position 2−8 nt) and overall mismatch ≤2.

**Gene family analyses.** Gene family analysis was performed using OrthoMCL[64]. The protein-coding genes of *L. vannamei* and nine other arthropods (downloaded from NCBI: www.ncbi.nlm.nih.gov), including *Strigamia maritima*, *Ixodes scapularis*, *Tetranychus urticae*, *P. hawaiensis*, *D. pulex*, *Acyrthosiphon pisum*, *L. migratoria*, *B. mori*, and *D. melanogaster*, were aligned to each other using the BLASTP program. Similarity information from the pairwise sequence alignments was used as distance parameters for gene family clustering. Species-specific genes were those that had no homologs in the other species used in this analysis. Orphan genes are a group of genes identified from species-specific genes that have no homologs to any genes from NCBI Nr database. However, these orphan genes are supported by transcriptome data.

**Phylogenetic tree construction.** A phylogenetic tree was constructed using the single-copy homologous genes from *D. pulex*, *P. hawaiensis*, *D. melanogaster*, *A. gambiae*, *L. migratoria*, *B. mori*, *Tribolium castaneum*, *Apis mellifera*, *Zootermopsis nevadensis*, *P. humanus*, *A. pisum*, *I. scapularis*, *T. urticae*, and *S. maritima*, with the RAxML software that is based on the maximum likelihood method[65]. Briefly, the homologous genes from the genomes of these species were clustered together using the OrthoMCL software. Each of these clusters was then filtered to obtain a total of 88 single-copy orthologs using our in-house Python script (find_multiple_snp.py). MUSCLE (http://www.drive5.com/muscle) was used for sequence alignment, at its default setting. Positions with gaps and missing data were trimmed using an in-house Python script (allfasta2snp.py). The final dataset contained 19,161 amino acids and was used to construct the phylogenetic tree with RAxML[66] under the JTT matrix-based model[67]. Initial trees for the heuristic search were obtained automatically by applying neighbor-joining and BioNJ algorithms to a matrix of pairwise distances estimated using a JTT model. The best tree had a log likelihood of −3308.02, and was used as an input tree for divergence time estimation by Bayesian relaxed molecular clock approaches implemented in MCMCTREE from the PAML package[68]. CODEML from the PAML package was implemented to estimate the neutral substitution rate per year among species. MCMCTREE was then used to calculate gradient vector ($g$) and Hessian matrix ($H$). Fossil calibrations were used as priors for the divergence time estimation[69]. Cauchy distributions were used with default parameters "$p = 0.1, c = 1$". The MCMCTREE is running twice each for 1,000,000 generations with the sample frequency of 50 and a burn-in phase of 50,000 iterations. Tracer (http://beast.bio.ed.ac.uk/) was used to assess chain convergence and adequate effective sample sizes of all parameters. The resulting time-calibrated tree was visualized by using MEGA7[66].

**Positive selection.** To assess the contribution of natural selection on the single-copy orthologs in the *L. vannamei* genome, the ratios ($\omega$) of nonsynonymous substitution per nonsynonymous site (dN) to synonymous substitution per synonymous site (dS) were calculated using the software package PAML v4.48a[68]. The homologs were aligned using MUSCLE. The branch-site model was used to detect positive selection along the foreground branch. Likelihood ratio tests were applied to test significant differences between the alterative and null models.

**Regulation of MIH-mediated ecdysone synthesis.** To examine how the cholesterol level and photoperiod regulate molting via MIH, we conducted SREBP gene RNAi, MIH genes RNAi, opsin genes RNAi, Lipitor injection, and cholesterol injection experiments, using dsEGFP RNAi and alcohol/phosphate buffer saline (PBS) injection as negative control treatments. The experiments used healthy shrimps with a body length of 9.55 ± 0.25 cm and a body weight of 11.20 ± 1.36 g, which were cultured and acclimated in fiberglass tanks with aerated circulating seawater for 2 days before the experiment.

To synthesize dsRNA for RNAi experiment, DNA templates were amplified from the eyestalk cDNA sample. The primer pair, dsEGFP-F and dsEGFP-R (Supplementary Table 25), was used to amplify a 289 bp DNA fragment of the EGFP gene from the pEGFP-N1 plasmid. The dsRNA of *SREBP* containing a fragment of 754 bp located at the 3′ end of the coding region, and the dsRNA of *MIH1036-6* containing a fragment of 377 bp covering the whole coding region, were amplified using the same method with primers in Supplementary Table 25. The dsRNA of LW opsin gene (*LW-1*) were amplified using two pairs of primers designed using the conserved regions of the genes (Supplementary Table 26). The PCR product (~500 bp) was generated using a T7 promoter anchor attached to the two primers used to amplify the product. The PCR products were purified using Gel Extraction Kit (OMEGA, Japan). dsRNA was synthesized with TranscriptAid T7 High Yield Transcription Kit (Thermo Fisher Scientific, USA), in accordance with the manufacturer's protocol.

We optimized the effective silencing dose of each dsRNA by injecting three different dosages, namely 1, 2, and 4 μg, of dsMIH1036-6 or dsSREBP or dsLW-1 or dsEGFP (as control) into healthy shrimps at the intermolt stage (C). Five individuals were tested for each dosage for dsMIH1036-6 and dsSREBP, while each dosage of dsLW-1 was tested on three individuals. At 48 h after dsRNA injection, the eyestalks of the shrimps in each group were collected for RNA extraction. Expression levels of each gene were determined by qPCR. The primers used for qPCR are listed in Supplementary Tables 25 and 26. Primers 18S-qF and 18S-qR were used to detect the expression of the internal reference gene, 18S rRNA. The specificity of the qPCR product was validated by melting-curve analysis performed at the end of each qPCR reaction. The optimal silencing doses were used in subsequent experiments.

To test the effect of SREBP on *MIH* expression, 20 individuals at the postmolt (P) stage were divided into two groups. In the experimental group, each individual was injected with 1 μg of dsSREBP, while individuals in the control group were injected with 1 μg of dsEGFP each. To test the effect of high cholesterol level on *MIH* expression, 20 individuals at the postmolt (P) stage were divided into two groups. Each individual in the experimental group was injected with 400 μg cholesterol dissolved in 100% alcohol. The same volume of alcohol was injected into individuals in the control group. For both experiments, the eyestalks of all shrimps in each group were collected at 48 h after injection and total RNA was extracted from different samples using RNAiso Plus reagent (TaKaRa, Japan). Expression levels of three MIH genes were estimated by qPCR, as described above.

To test the effect of opsin on *MIH* expression, shrimps at the same molting stage (D0-D1) were divided into six groups: three dsLW-1 groups ($n = 20$) and three groups for dsEGFP injection ($n = 20$) as controls. The concentration of dsRNA was 1 μg for all groups. We examined the occurrence of molting three times a day after RNAi (at 08:00, 14:00 and 20:00). After silencing for 48, 72, and 96 h, three shrimps from each group were selected to determine the expressions of MIH and three LW opsin genes by qPCR.

During the aforementioned *opsin* RNAi experiment, we examined the occurrence of molting three times a day (at 08:00, 14:00, and 20:00). At 9 days after silencing, we administered a second injection to ensure the effects of silencing. The observation lasted for 18 days.

To examine the effect of *MIH* silencing on molting, 20 individuals at the intermolt stage (C) were divided into two groups. Each individual in the experimental group was injected with 4 μg MIH1036-6 dsRNA, and each individual in the control group was injected with 4 μg EGFP dsRNA. The tails of all the individuals in different groups were dissected for observation of the molting stage under a microscope at 48 h after dsRNA injection. To test the effect of low cholesterol level on molting, 20 individuals at the early premolt stage (D0–D2) were divided into two groups. Each individual in the experimental group was injected with 5 μg atorvastatin calcium (Lipitor) predissolved in 20 μl PBS (with 2.50% DMSO), and each shrimp in the control group was injected with 20 μl PBS (with 2.50% DMSO). The uropods of all the individuals in the two groups were dissected for observation of the molting stage under a microscope at 48 h after injection.

**Resequencing and SNP identification.** A total of 22 shrimp individuals were selected for genome resequencing. These included eight wild individuals collected from Baja California Sur, Mexico, six cultured individuals selected for four generations in Ecuador, and eight individuals from four commercial breeding lines in China, the USA and Thailand (Supplementary Table 20), which had all been subjected to selective breeding for over 20 generations. The raw reads were mapped to the reference genome using BWA v0.7.12 [70]. The PCR duplicates of each of the samples were removed with Picard Mark Duplicates. Reads around INDELs from the BWA alignment were realigned using the IndelRealiger option in the Genome Analysis Toolkit (GATK). The HaplotypeCaller of the Genome Analysis Toolkit

GATK was used to construct the general variant calling file (gVCF) for every individual. The individual gVCF files were combined by GenotypeGVCFs function to form a single variant calling file, which includes all the sites. A strict filtering of the SNP calls was carried out using the guidelines given by the Broad Institute using the options: "FS > 60.0 || MQ < 40.0 || QD < 2.0 || SOR > 3.0". SNPs with minor allele frequency (MAFs) less than 5% were discarded. The missing rate of a genotype for population was 10%. Variants were annotated with the SNPeff annotation program to identify the synonymous and nonsynonymous mutations.

**Identification of candidate genes under breeding selection**. A phylogenetic tree of the 22 individuals was constructed using filtered SNP by the SNPhylo software (parameters: -l 0.6, -m 0.05). To identify the genomic regions significantly affected by artificial breeding, we calculated pairwise nucleotide diversity ($\pi_{wild}/\pi_{breeding}$) and divergence index ($F_{ST}$) between the wild and cultured populations using a sliding-window approach (100 Kb windows with 20 Kb increments) using VCFtools (http://vcftools.sourceforge.net/). The genomic regions with top 1% of rank level values for empirical percentiles detected by both methods were identified as potential selective sweeps. GO and KEGG enrichment of genes from the selective sweeps were performed using KOBAS (http://kobas.cbi.pku.edu.cn/).

**Code availability**. The associated Perl and Python scripts can be viewed and downloaded free at the github repository: https://github.com/jianbone/L_vannamei_genome. The software used for data analyses include: Trimmomatic v.0.35, BluePippin v6.31, Jellyfish-1.2.8, wtdbg-1.2.8, SMRT Analysis v2.3.0, BWA v0.7.12, SSPACE 3.0, Bowtie 2.3.4.3, CEGMA v3, RepeatModeler Open-1.0, RepeatMasker 4.0.5, Trinity v2.6.5, TopHat v1.2.1, Cufflinks v2.2.1, Exonerate v2.2.0, Augustus v2.5.5, Tophat v2.1.1, EvidenceModeler (EVM) r03062010, InterProScan 5, HMMER-3.0a1, miRDeep2, OrthoMCL v1.4, RAxML Workbench 1.0, MUSCLE 3.8.31, PAML v4.48a, MEGA7, Genome Analysis Toolkit (GATK) 3.6, SNPeff LGPLv3,VCFtools 1.0.0.0, KOBAS 3.0.

## Data availability

Genome sequences data that support the findings of this study have been deposited in NCBI GenBank with the accession codes of QCYY00000000. RNA-Seq data were used for annotation and biological analyses: NCBI SRA SRR1460493 −SRR1460495 (https://www.ncbi.nlm.nih.gov/Traces/study/?acc=SRP043546), SRR1460504−SRR1460505 (https://www.ncbi.nlm.nih.gov/Traces/study/?acc=SRP043546), SRX1098368−SRX1098375 (https://www.ncbi.nlm.nih.gov/Traces/study/?acc=SRP061180). The genome sequences can also be downloaded from the genome database of *L. vannamei* (http://www.shrimpbase.net/vannamei.html), where BLAST searching and gene structure visualization are also available. A reporting summary for this article is available as a Supplementary Information file. The source data underlying Figs. 1, 2, and 6b are provided as a Source Data file.

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

## Acknowledgements

We acknowledge financial support from the State 863 High Technology R&D Project of China (2012AA10A404 to J.X.), the National Natural Science Foundation of China (31830100 to F.L. and 41876167 to J.Y.), the grants from Qingdao National Laboratory for Marine Science and Technology (MS2017NO04 to F.L.), The Senior User Project of RV KEXUE (KEXUE2018G19 to F.L. and X.Z.), the China Agriculture Research system-48 (CARS-48 to F.L.) and the Hong Kong Collaborative Research Fund (CRF) project (No. C4042-14G to K.H.C). We acknowledge the support from High Performance Computing Center, Institute of Oceanology, CAS. We would like to express our gratitude to Prof. Jun Yu, Prof. Songnian Hu, Prof. Jingfa Xiao and Dr. Tongwu Zhang (Beijing Institute of Genomics, Chinese Academy of Sciences), Prof. Ximing Guo (Rutgers University), Dr. Ruiqiang Li, Dr. Zhi Jiang and Dr. Jinbo Zhang (Novogene Bioinformatics Institute), Dr. Junyi Wang and Dr. Dongliang Zhan (Hangzhou 1Gene Ltd), for their support of the shrimp genome project, genome sequencing and assembly. We appreciate Dr. Hao Huang (Hainan Guangtai Ocean Breeding Co., Ltd) for the help with shrimp materials and Dr. Yang Zhang and Dr. Han Cao (Bionano Genomics) help for the construction of optical map. We thank Cui Zhao, Jiankai Wei, Xiaoqing Sun, Jingwen Liu, Jiangli Du, Mingzhe Sun and Yan Zhang for help with DNA, RNA extraction, and data analysis. We are grateful to Prof. Li Sun, Prof. Linsheng Song, Prof. Jinsheng Sun, Prof. Zhaoxia Cui, Prof. Baozhong Liu and Dr. Pin Huan for helpful discussions. We thank other faculty and staff at the Institute of Oceanology, Chinese Academy of Science who contributed to the shrimp genome project. We thank Dr. Jing Qin for revisions of the manuscript. We appreciate Mr. Huawei Zhang for shrimp photography.

## Author contributions

J.X., F.L., Lei Wang and Xiaojun Zhang initiated, managed, and drove the shrimp genome sequencing project. Xiaojun Zhang, C.Z., K.Y., and J.K. collected the animal material. Xiaojun Zhang, Y.S., S.L., Y.G., J.Y., and Y.Y. prepared DNA sequencing and RNA-Seq analysis. J.Y., Y.S., J.R., B.L., X.W., W.L., X. Zhu, and Long Wang performed genome assembly, gene annotation, genome structure analyses, and phylogenetic analyses. Y.G., Xiaojun Zhang, and J.Y. conducted the environment adaptation analysis. S.L., Xiaojun Zhang, Y.G., K.Y.M., and J.Y. conducted the molting regulation analysis. Y.Y., Q.W., and Y.S. conducted the population genetics analysis. S.L, Y.G., X.L., Xiaoxi Zhang, J.Z., and S.J. performed the validation experiments. J.Y., Y.S., and C.L. submitted the genome data. C.L. performed the ncRNA analysis and constructed the shrimp genome database. J.X., F.L., Xiaojun Zhang, J.Y., S.L., Y.G., Y.Y., Y.S., and C.L. wrote the manuscript and additional supplementary files. K.H.C., K.Y.M., J.C., H.Z., Z.L., P.X., A.S., P.S., J.R., A.W., B.L., and Lei Wang revised the manuscript. All authors read and approved the final manuscript.

## Additional information

**Competing interests:** The authors declare no competing interests.

