## [Peer Review File · Nature Communications]

Reviewer #1 (Remarks to the Author):

This is comprehensive analysis of high quality genome sequence of an important decapod crustacean species. While none of the analysis in this paper is particularly striking or exciting the key contribution of this paper is that it now makes a high quality genome sequence available to the community. In addition to the a reference genome the paper compares wild and selectively bred *L. vannamei*, clearly showing a signal of selective breeding and reduction in genome wide polymorphism. It always challenging to make genome papers exciting when the main goal is to make the genome available, but the authors make a reasonable attempt to relate the genome to important aspects of *L. vannamei* biology.

Major comments on the paper.

- 1) Given the importance of pathogens and immune responses in aquaculture of this important crop species I would expect a comprehensive comparative analysis of immune components. If this has been undertaken elsewhere recently this should be cited or if not should be a part of this paper.
- 2) The presentation and textual explanation of Figure 4 (in relation to molting biology) is poor and difficult to follow. The significance of the data presented needs to be clearly explained in relation to the biology of the animal.

Minor comments

- 1) Throughout the authors perform comparative analysis with a number of other crustacean genomes (e.g *Daphnia*, *Parhyale* other decapods etc). The cognate genome papers must be cited in the main text.
- 2) The authors provide data showing the efficacy of their RNAi treatments in this study. It seems that the control condition has a significant effect on unnamed control gene expression. The authors should comment on the efficiency and efficacy of RNAi in this species to put the strength of their data in the correct context.

Reviewer #2 (Remarks to the Author):

General comments:

The authors present the first good quality genome assembly of a Penaeid shrimp, and also provide an impressive array of additional analyses, experiments and biological context. Overall, the manuscript is well structured and well written. Of highest interest to the scientific community will be the genome itself, which is a valuable resource. The genome assembly itself is well done and sufficiently described. The additional analyses and biological context are scientifically sound and directly tie in with the genome, which is the core of the manuscript, but are then taken several steps further. This results in these extra sections being quite extensive, and while they are generally well described, the short nature of this manuscript limits the information and discussion that can be provided. For example, the section covering molting looks at a genes and gene complexes, found through genome annotation, which are involved in the regulation of molting. However, this is then taken further and the authors provide information on an experiment where molting was modulated through the injection of varying compounds. This in itself could probably be a publication in its own right. This is also noticeable in the number of panels in each of the five figures, which are very information dense. While all figures are visually appealing, it is often difficult to see details. For some of them zooming in helps, but this is not always the case (see specific comments below).

It would also be useful for the scientific community if more of the code used for this publication was made publicly available? For example, the gene family analysis or phylogenetic tree construction using an in-house perl script. As is, it would not be possible to exactly replicate all analyses based on the given information.

Some specific comments:

It would be useful if the abstract contained some of the assembly metrics (N50, # contigs).

Similarly, it would be useful to have # contigs mentioned in text in the Result section (lines 79 to 93), rather than having them in the supplementary material.

Line 112: change “that identified by” with “that was identified”

Line 119 to 120: this part doesn't fit the first half of the sentence, and is also redundant with line 111: “but the 120 (AACCT)_n type SSRs are longer than other SSRs.”

Line 126: change “dramatically higher than that in” to “dramatically higher than in”

Line 177: delete “for”

Line 181: change to “might have prompted”

Line 183 to 185: Unclear sentence. Are the “two predominate genomic characters” under natural selection? And how do they “provide genetic architecture for the evolution of the ancestor”?

Line 234: replace “tens of thousands” with value in bracket

It is not quite clear to me what the difference is between species-specific and orphaned genes.

Figure 1 is very crowded. If there is enough space available, it would be better to split some of the components off and make them larger. Panel c, while visually appealing, could probably be moved into the supplementary material. I’m not sure if it makes sense to represent expansion and contractions of gene families as a pie chart. What would be 100% (i.e. a full circle), if there are both gains and losses in genes in the same pie chart? Not sure if panel g adds much.

The same goes for Figures 2 and 3 and 5. Especially in Fig 3 where panel e is very, very small and blurry.

Supplementary Figure 3 would be easier readable as a table.

Supplementary tables 11 and 12 (and probably others) should contain references from where the data not generated as part of this manuscript was obtained.

Fig. 5a: What are the source populations of the breeding shrimp? Could the differences seen between the wild shrimp from Mexico and the breeding shrimp be due to different stocks, rather than breeding?

Line 950-951:Methods: how were positions containing gaps and missing data eliminated?

Reviewer #3 (Remarks to the Author):

Dear Editor,

Thank you for the opportunity to review the manuscript titled: “Decoding the complex penaeid shrimp genome provides insights into benthic adaptation, frequent molting and breeding impacts”.

In this manuscript, the Authors describe several aspects of the genome of the whiteleg shrimp *Litopenaeus vannamei*, the commercially most important shrimp world-wide. The researchers provide insights into the shrimp benthic habitat adaptations through examining the expansion of relevant gene families. Additionally, the Authors provide mechanistic evidence for the opsins and SREBP control over molting through promoter analysis and gene silencing experiments. This manuscript represents an important milestone for decapod crustacean researchers, given the scarcity of high quality assembled genomes available for this taxonomic group and the exceptionally low level of genes annotated when compared with *Daphnia* and insects. The level of work is very comprehensive and outstanding, although I suggest major revisions, mainly to improve the readability and access to information for the broad readership of Nature Communications.

General comments:

The manuscript reads quite well but it would be advised to re-structure and prune it as detailed below.

It is hard to follow the experimental designs and the reference to the supplementary files is in some cases misleading. For example, when referring to the primers used for gene silencing, the SREBP primers do not appear in Table 31. The region of the gene used to design dsRNA and the gene accession number are not available. The caption of Supplementary Figure 24 describes it shows silencing of either SREBP or MIH, whereas in the main text (lines 403-404) it discusses SREBP silencing effect on MIH.

In several occasions the results section includes discussion (see for example lines 186-196, 424-431). I suggest that the Authors consider combining the results and discussion.

The discussion can be easily amalgamated into the results section and instead a conclusion paragraph can be added to succinctly describe the key findings of the manuscript. As it is, it is quite fragmented and requires the reader to read back and forth to gather key information. As an example, the PTH and PG are described for the first time late in the manuscript (lines 556-558) and are related to the results and discussion part that is provided much earlier.

Language editing is required. There are many typos and grammatical errors throughout the text. Some are elaborated below.

The exceptionally high level of SSRs is flagged as a hurdle for sequencing but was not addressed in the context of the biology. What does this represent? High mutation rate? High level of interaction with viruses and horizontal transmission? High transposable elements activity? The result section starts with the explanation of the difficulties in the sequencing effort. It is highly appreciated, but it detracts from the biology of the research. If the authors wish to discuss the difficulty, they could consider adding recommendations for future research how to circumvent these difficulties. In addition, it is mentioned that there was an initial assessment of the level of repetitiveness in the genome, but the details of this initial assessment were not provided. Additionally, when it comes to level of repetitive elements in the genome, as well as complexity in terms of polyploidy, it is known that many plants have much higher complexity in their genomes and this is expected to be discussed as a highly relevant comparison. A relevant example is the rubber tree genome with 71% repetitive content, far higher than that of the shrimp.

Following the paragraph describing the high SSR rate the Authors then describe expansion in gene families and the molt regulation mechanisms. It would make more sense to discuss the effect of breeding with the level of SNPs identified, prior to describing the gene families' expansion. This would make a nicer flow and will put two parts that are more related one next to another. If the Author accept this change, the title, abstract, introduction and discussion should be revised accordingly as well.

Specific comments:

Line 14 (as well as lines 33-34): 'the second largest...' this statement is not correct. Chelicerata have more identified species than crustacea. Perhaps rephrasing to highlight that together with the largest arthropod group of hexapoda, crustacea form the pancrustacea clade. Not only that this clade is the vast majority of species, but it also emphasizes that hexapoda are land-adapted derived crustaceans. This phrasing positions crustaceans as the ancestors of the dominating subphylum on the planet.

Line 41: 'over years' should be 'over the years'.

Lines 49-51: this statement should be refined. Non-metabolous and some hemimetabolous insects also molt throughout their lives, unlike holometabolous insects.

Line 52: what is the shrimp life span? What is the frequency of the 50 molts?

Line 53: more frequent than other arthropods. Please add references.

Line 61: mot important or most commercially important?

Line 64: please add 'a' between 'with' and 'number'.

Line 112: add 'was' before 'identified'.

Figure captions could be judiciously pruned. These are very lengthy and overly detailed. Some of the information could be transferred to supplementary files.

Lines 276-280: the cholesterol transport is also key for producing ecdysteroids and sesquiterpenoids. This should be mentioned here (even if it is described later in line 397). Given that crustaceans can't synthesize cholesterol and rely on its dietary availability, it would be expected that the genes responsible for cholesterol transport will be highly expressed in the digestive system.

Lines 291-294: it would be interesting to discuss winged insects in the context of locomotion, given that they derived from crustaceans and they also hold the record for the fastest contracting muscles (wings). The wings derived from ancestral gills and it would therefore make sense to explore the evolutionary trajectory for this divergence.

Line 302: 'with retinal...' is there a word missing?

Lines 314-337: this subsection is hard for to follow for the generalist reader. It is filled with terms that are not elaborated anywhere in the text (for example R&R1, D2, D4) and could be simplified. Additionally, the ecdysone cascade is presented with very little background and provides the reader with little information. For example, there is no reference to the ecdysone synthesis pathway, the Y organ, the fact that there are multiple nuclear receptors (including several ecdysone receptor isoforms), some of which are activated/inhibited by the ecdysone peak. It calls for a more elaborated and structured writing or at least more references to reviews of these processes. Additionally, it is not clear how Br-C is linked to ecdysone. More details are required and should have been provided in the introduction as well.

Lines 554-564: this reads like introduction rather than discussion. It re-iterates some concepts that were introduced earlier and then provides new information that would suite better in the introduction.

Comments from Reviewer #1:

This is comprehensive analysis of high quality genome sequence of an important decapod crustacean species. While none of the analysis in this paper is particularly striking or exciting the key contribution of this paper is that it now makes a high quality genome sequence available to the community. In addition to the a reference genome the paper compares wild and selectively bred *L. vannamei*, clearly showing a signal of selective breeding and reduction in genome wide polymorphism. It always challenging to make genome papers exciting when the main goal is to make the genome available, but the authors make a reasonable attempt to relate the genome to important aspects of *L. vannamei* biology.

Major comments on the paper.

1) Given the importance of pathogens and immune responses in aquaculture of this important crop species I would expect a comprehensive comparative analysis of immune components. If this has been undertaken elsewhere recently this should be cited or if not should be a part of this paper.

Response:

Yes, decoding the genome of *L. vannamei* certainly provides a lot of information about the immune components in crustaceans, which are very important to the aquaculture species. We are now conducting a comprehensive comparative analysis of immune components. Considering that one of the focuses of this manuscript is molting biology in penaeid shrimp, we added data on the immune protection during molting following the section on “Regulation of frequent molting” in Results and Discussion (Lines 480-522).

2) The presentation and textual explanation of Figure 4 (in relation to molting biology) is poor and difficult to follow. The significance of the data presented needs to be clearly explained in relation to the biology of the animal.

Response:

We have modified the presentation and textual explanation of the Figure 4 (Figure 5 in the revised manuscript) (Lines 417-478) and hope that it would be clearer.

Minor comments

1) Throughout the authors perform comparative analysis with a number of other crustacean genomes (e.g *Daphnia*, *Parhyale* other decapods etc). The cognate genome papers must be cited in the main text.

Response:

We incorporated the relevant references of other crustacean genomes in the Introduction section (Lines 75-77).

2) The authors provide data showing the efficacy of their RNAi treatments in this

study. It seems that the control condition has a significant effect on unnamed control gene expression. The authors should comment on the efficiency and efficacy of RNAi in this species to put the strength of their data in the correct context.

Response:

Sorry for the causing confusion on this aspect. We have revised the figure to show the dsRNA-mediated silencing efficiency of Opsin genes at different times (Supplementary Figure 25) following your suggestion; the gene expression of control (dsEGFP injection) has been normalized.

RNAi in crustaceans including shrimp has been widely used and proved to be an effective approach to study the gene function. Following are several examples:

1. Sagi A, Manor R, Ventura T. Gene silencing in crustaceans: from basic research to biotechnologies. *Genes*, 2013, 4:620-645.
2. Itsathitphaisarn O, Thitamadee S, Weerachayanukul W, Sritunyalucksana K. Potential of RNAi applications to control viral diseases of farmed shrimp. *Journal of Invertebrate Pathology*, 2017, 147:76-85.
3. Priya T A J, Li F, Zhang J, Yang C, Xiang J. Molecular characterization of an ecdysone inducible gene E75 of Chinese shrimp *Fenneropenaeus chinensis* and elucidation of its role in molting by RNA interference. *Comparative Biochemistry and Physiology Part B: Biochemistry and Molecular Biology*, 2010, 156(3): 149-157.

Comments from Reviewer #2:

General comments:

The authors present the first good quality genome assembly of a Penaeid shrimp, and also provide an impressive array of additional analyses, experiments and biological context. Overall, the manuscript is well structured and well written. Of highest interest to the scientific community will be the genome itself, which is a valuable resource. The genome assembly itself is well done and sufficiently described. The additional analyses and biological context are scientifically sound and directly tie in with the genome, which is the core of the manuscript, but are then taken several steps further. This results in these extra sections being quite extensive, and while they are generally well described, the short nature of this manuscript limits the information and discussion that can be provided. For example, the section covering molting looks at a genes and gene complexes, found through genome annotation, which are involved in the regulation of molting. However, this is then taken further and the authors provide information on an experiment where molting was modulated through the injection of varying compounds. This in itself could probably be a publication in its own right. This is also noticeable in the number of panels in each of the five figures, which are very information dense. While all figures are visually appealing, it is often difficult to see details. For some of them zooming in helps, but this is not always the case (see specific comments below).

It would also be useful for the scientific community if more of the code used for this publication was made publicly available? For example, the gene family analysis or phylogenetic tree construction using an in-house perl script. As is, it would not be

possible to exactly replicate all analyses based on the given information.

Response:

Many thanks for the comments. We have submitted the relevant Perl and Python scripts at the github repository: https://github.com/jianbone/L_vannamei_genome.

Some specific comments:

It would be useful if the abstract contained some of the assembly metrics (N50, # contigs).

Response:

We added the assembly information of scaffold N50 in the abstract (Lines 15-16).

Similarly, it would be useful to have # contigs mentioned in text in the Result section (ss 79 to 93), rather than having them in the supplementary material.

Response:

We added the scaffold number in the Result & Discussion section (Line 95). We also added a table (Table 1) that describes the details of the assembly and genomic characteristics.

Line 112: change “that identified by” with “that was identified”

Response:

This sentence has been amended as “a telomere component identified by fluorescence in situ hybridization” (Lines 126-127).

Line 119 to 120: this part doesn't fit the first half of the sentence, and is also redundant with line 111: “but the 120 (AACCT)_n type SSRs are longer than other SSRs.”

Response:

We revised this sentence as follows (Lines 126-128): “(AACCT)_n, a telomere component identified by fluorescence in situ hybridization (Fig. 1c), is longer than other SSRs. In fact, the longest SSR, with a length of 13,769 bp, belongs to the (AACCT)_n type, suggesting its role in chromosome stability (Supplementary Fig. 6).”

Line 126: change “dramatically higher than that in” to “dramatically higher than in”

Response:

Changed.

Line 177: delete “for”

Response:

Deleted.

Line 181: change to “might have prompted”

Response:

Changed.

Line 183 to 185: Unclear sentence. Are the “two predominate genomic characters” under natural selection? And how do they “provide genetic architecture for the evolution of the ancestor”?

Response:

This part has been revised to: “We found three prominent genome characteristics from *L. vannamei* that might underlie the rapid evolution of the penaeid shrimp, namely, the existence of abundant SSRs, a high proportion of species-specific genes, and extensive tandem duplication of genes.” (Line 177-180)

Line 234: replace “tens of thousands” with value in bracket

Response:

Replaced.

It is not quite clear to me what the difference is between species-specific and orphaned genes.

Response:

Species-specific genes were collected from the gene family analysis, which compared the full protein-coding genes against the genes from other crustaceans with genomes available. As limited crustacean genomes are available, the species-specific genes of crustaceans might be overestimated. Orphan genes are a group of genes collected from species-specific genes, which did not show any homologues to the genes from other genome-available species, and showed no homologues to the genes from NCBI Nr database either. Furthermore, orphan genes may be independent *de novo* originated as described in the Results and Discussion section (Lines 205-207). Besides, we also added the relevant description of species-specific genes and orphan genes at the Methods section (Lines 751-755).

Figure 1 is very crowded. If there is enough space available, it would be better to split some of the components off and make them larger. Panel c, while visually appealing, could probably be moved into the supplementary material. I’m not sure if it makes sense to represent expansion and contractions of gene families as a pie chart. What would be 100% (i.e. a full circle), if there are both gains and losses in genes in the same pie chart? Not sure if panel g adds much.

Response:

Thank you for your suggestion. We divided Figure 1 into two figures. The original panels c and g provide important information of the telomere and orphan genes, so we kept them in the two figures. The pie charts in Panel e are redundant and are removed.

The same goes for Figures 2 and 3 and 5. Especially in Fig 3 where panel e is very, very small and blurry.

Response:

We revised and rearranged these figures to make them more clearly.

Supplementary Figure 3 would be easier readable as a table.

Response:

We changed Supplementary Figure 3 into a table (Supplementary Table 5).

Supplementary tables 11 and 12 (and probably others) should contain references from where the data not generated as part of this manuscript was obtained.

Response:

We have added references in Supplementary tables 12 and 13.

Fig. 5a: What are the source populations of the breeding shrimp? Could the differences seen between the wild shrimp from Mexico and the breeding shrimp be due to different stocks, rather than breeding?

Response:

The sources of the commercial lines from USA, China and Thailand include a mixture of domesticated lines and wild shrimp of several origins, which includes the wild population in Mexico. The population of Ecuador breeding shrimp was from the wild shrimp near the coast of Ecuador. All the four breeding lines were combined as breeding populations, so the source of breeding populations and wild population from Mexico was related. Besides, we used two methods including pairwise nucleotide diversity ($\pi_{wild}/\pi_{breeding}$) and divergence index (F_{ST}) to identify the genomic region under selective pressure. Those regions showing difference between stocks are eliminated and only the highly selected regions are kept. Interestingly, although large number of breeding shrimps with different origins were analyzed, the genetic diversity of these breeding shrimp was low, indicating the selective pressure on the genome.

Line 950-951: Methods: how were positions containing gaps and missing data eliminated?

Response:

We eliminated the gaps and missing data by using an in-house Python script (allfasta2snp.py) (Line 766-767), which was submitted on the github repository: https://github.com/jianbone/L_vannamei_genome (line 899-901, Methods, Data availability section).

Responses to Reviewer #3:

Thank you for the opportunity to review the manuscript titled: “Decoding the complex penaeid shrimp genome provides insights into benthic adaptation, frequent molting and breeding impacts”.

In this manuscript, the Authors describe several aspects of the genome of the whiteleg shrimp *Litopenaeus vannamei*, the commercially most important shrimp world-wide. The researchers provide insights into the shrimp benthic habitat adaptations through examining the expansion of relevant gene families. Additionally,

the Authors provide mechanistic evidence for the opsins and SREBP control over molting through promoter analysis and gene silencing experiments. This manuscript represents an important milestone for decapod crustacean researchers, given the scarcity of high quality assembled genomes available for this taxonomic group and the exceptionally low level of genes annotated when compared with *Daphnia* and insects. The level of work is very comprehensive and outstanding, although I suggest major revisions, mainly to improve the readability and access to information for the broad readership of Nature Communications.

General comments:

The manuscript reads quite well but it would be advised to re-structure and prune it as detailed below.

It is hard to follow the experimental designs and the reference to the supplementary files is in some cases misleading. For example, when referring to the primers used for gene silencing, the SREBP primers do not appear in Table 31. The region of the gene used to design dsRNA and the gene accession number are not available. The caption of Supplementary Figure 24 describes it shows silencing of either SREBP or MIH, whereas in the main text (lines 403-404) it discusses SREBP silencing effect on MIH.

Response:

Thanks for the comments. We examined the relevant experimental design carefully. We added the information of primers (Supplementary Table 32) and the region used for dsRNA design (Lines 803-804) for SREBP gene in the revised manuscript. Supplementary Figure 24 showed the silencing efficiency of SREBP and MIH. In the main text, we made a mistake in discussing the results. We have now modified it into "SREBP silencing in eyestalk (Supplementary Fig. 24) significantly decreased the expression levels of all detected MIH genes by about 95% (Fig. 5b)" in the revised manuscript (Lines 433-434).

In several occasions the results section includes discussion (see for example lines 186-196, 424-431). I suggest that the Authors consider combining the results and discussion.

Response:

We appreciate the reviewer's suggestion. We have now combined the Results and Discussion into one section, which is followed by a section on Summary and Discussions.

The discussion can be easily amalgamated into the results section and instead a conclusion paragraph can be added to succinctly describe the key findings of the manuscript. As it is, it is quite fragmented and requires the reader to read back and forth to gather key information. As an example, the PTTH and PG are described for the first time late in the manuscript (Lines 556-558) and are related to the results and discussion part that is provided much earlier.

Response:

As aforementioned, we have now combined the Results and Discussion into one

section, which is followed by a section on Summary and Discussions.

Language editing is required. There are many typos and grammatical errors throughout the text. Some are elaborated below.

Response:

Thanks for reviewer's suggestion. We have sought the assistance of an English editor, Dr. David Wilmshurst in English editing.

The exceptionally high level of SSRs is flagged as a hurdle for sequencing but was not addressed in the context of the biology. What does this represent? High mutation rate? High level of interaction with viruses and horizontal transmission? High transposable elements activity? The result section starts with the explanation of the difficulties in the sequencing effort. It is highly appreciated, but it detracts from the biology of the research. If the authors wish to discuss the difficulty, they could consider adding recommendations for future research how to circumvent these difficulties. In addition, it is mentioned that there was an initial assessment of the level of repetitiveness in the genome, but the details of this initial assessment were not provided. Additionally, when it comes to level of repetitive elements in the genome, as well as complexity in terms of polyploidy, it is known that many plants have much higher complexity in their genomes and this is expected to be discussed as a highly relevant comparison. A relevant example is the rubber tree genome with 71% repetitive content, far higher than that of the shrimp.

Response:

The biology significance of SSRs is generally related to sustaining genome structure and regulating gene expression. The telomere sequence- (AACCT)_n type SSR was much longer than most SSRs, and thus they might be responsible for the stability of chromosomes. SSRs have been identified to play critical roles in regulating the genome plasticity (including DNA recombination and replication) and gene expression. Thus, the significant expansion of SSRs might provide a unique genetic architecture for the shrimp's adaptive evolution. However, further studies on SSRs are needed to enhance the understanding of their biological functions. We added the relevant discussion in the section Results and Discussion (Lines 190-196).

The assessment of repetitive sequences and the comparison with other crustaceans were provided in the supplementary materials (Supplementary materials Tables 10, 12). We would like to compare repetitive elements with many other species. However, we only identified relative little transposable elements (TEs, 16.17%) in the genome of *L. vannamei*. SSR is a different kind of repetitive sequences from TEs, and it is also the major repetitive sequences in the genome of *L. vannamei* (23.93%). The percentage is the highest among the species for which the genomes are available (Fig. 1b, Supplementary materials Table 11). The percentage of SSRs in plant genomes are less than 3%.

Following the paragraph describing the high SSR rate the Authors then describe expansion in gene families and the molt regulation mechanisms. It would make more

sense to discuss the effect of breeding with the level of SNPs identified, prior to describing the gene families' expansion. This would make a nicer flow and will put two parts that are more related one next to another. If the Author accept this change, the title, abstract, introduction and discussion should be revised accordingly as well.

Response:

In this manuscript, the high abundant of SSR is a remarkable character of *L. vannamei* genome and the following part of the manuscript was also related to the genome character, so we put these information together. The SNPs identification was based on the genome resequencing analysis of 22 individuals from the wild population and breeding broodstocks. We think the flow from basic to applied studies in the original manuscript is ok so that we have not followed the reviewer's comment on rearranging the different parts.

Specific comments:

Line 14 (as well as lines 33-34): 'the second largest...' this statement is not correct. Chelicerata have more identified species than crustacea. Perhaps rephrasing to highlight that together with the largest arthropod group of hexapoda, crustacea form the pancrustacea clade. Not only that this clade is the vast majority of species, but it also emphasizes that hexapoda are land-adapted derived crustaceans. This phrasing positions crustaceans as the ancestors of the dominating subphylum on the planet.

Response:

Thanks for reviewer's excellent suggestion. We revised the first sentence to: "Crustacea, the subphylum of Arthropoda which dominates the aquatic environment, is of major importance in ecology and fisheries." (Lines 13-14) The second sentence has been revised to: "The two groups constitute the Pancrustacea clade, with Crustacea as an early divergent, paraphyletic assemblage with respect to the derived Hexapoda, the most successful group of organisms on land." (Lines 33-36)

Line 41: 'over years' should be 'over the years'.

Response:

Amended.

Lines 49-51: this statement should be refined. Non-metabolous and some hemimetabolous insects also molt throughout their lives, unlike holometabolous insects.

Response:

Thanks for reviewer's suggestion. We revised it as "Molting in crustaceans differs from that in most insects, where molting usually occurs at the larval stages for growth and metamorphosis. Molting occurs throughout the lifetime of crustaceans. For instance, the penaeid shrimp experiences about 50 molts during a lifetime⁶, a far higher number than in other arthropods" (Lines 51-55)

Line 52: what is the shrimp life span? What is the frequency of the 50 molts?

Response:

Generally, the life span of shrimps is about one year. The frequency and time between molts depend on different developmental stages. During the early larval stages, it usually experience 12 times molting for metamorphosis. During the grow-up stages, the juvenile shrimp molt more frequently than the older ones.

Line 53: more frequent than other arthropods. Please add references.

Response:

L. vannamei experiences 50 moltings during a lifetime on average, which is much more than in other arthropods; e.g., silkworms (four times) (Sarkar, et al, 2015), crabs (~18 times) (Chen et al., 2016), and locusts (five times) (Fuzeaubraesch et al., 1979). According to reviewer's suggestion, we have added the related references in the revised manuscript.

1. Sarkar B N, Sarmah M C, Ahmed S A, et al. Eri silkworm rearing on perennial host plants[J]. Indian Silk, 2015, 53(12):30-32.
2. Chen J, Xuzhou M A, Wang W, et al. The study of relationships between growth, molt and accumulated temperature of the Chinese mitten crab(*Eriocheir sinensis*)[J]. Journal of Shanghai Ocean University, 2016.
3. Fuzeaubraesch S, Coulon JF, David JC (1979) OCTOPAMINE LEVELS DURING THE MOLT CYCLE AND ADULT DEVELOPMENT IN THE MIGRATORY LOCUST, *LOCUSTA-MIGRATORIA*. *Experientia* 35: 1349-1350.

Line 61: not important or most commercially important?

Response:

Should be the most commercially important.

Line 64: please add 'a' between 'with' and 'number'.

Response:

Done.

Line 112: add 'was' before 'identified'.

Response:

Done.

Figure captions could be judiciously pruned. These are very lengthy and overly detailed. Some of the information could be transferred to supplementary files.

Response:

Thanks for reviewer's suggestion. In the revised manuscript, we shortened the legends for all figures.

Lines 276-280: the cholesterol transport is also key for producing ecdysteroids and sesquiterpenoids. This should be mentioned here (even if it is described later in line 397). Given that crustaceans can't synthesize cholesterol and rely on its dietary availability, it would be expected that the genes responsible for cholesterol transport

will be highly expressed in the digestive system.

Response:

In crustaceans, Crustacyanins are the main molecules responsible for cholesterol transport. As the reviewer speculated, many of these genes are highly expressed in the digestive system (intestine, stomach and hepatopancreas) (Supplementary Table 20).

Lines 291-294: it would be interesting to discuss winged insects in the context of locomotion, given that they derived from crustaceans and they also hold the record for the fastest contracting muscles (wings). The wings derived from ancestral gills and it would therefore make sense to explore the evolutionary trajectory for this divergence.

Response:

Thanks for the reviewer's excellent suggestion. The origin of insect wings and their evolutionary trajectory from the crustaceans are certainly a most interesting topic. Although insect wings are often used as an example of morphological novelty, the origin of insect wings remains not very clear (Clark-Hachtel and Tomoyasu, 2016). Recent investigations using the evolutionary developmental biology (evo-devo) approach suggested the origin of insect wings was diverse. We searched the hox genes in the genome of shrimp, and identified most of the hox genes (e.g. *lab*, *pb*, *Hox3*, *Dfd*, *Scr*, *ftz*, *Antp*, *Ubx*, *AbdA*, *AbdB*, *Msx*, *Mnx* and *Dll*). Among them, *Scr*, *Ubx*, *AbdA*, and other hox genes were reported to be related to insect wing development, but we did not find obvious hox gene cluster in the shrimp. We also found other wings related genes of insects in the shrimp genome, including: *vestigial (vg)*, *apterous (ap)*, *disheveled (dsh)*, *nubbin (nub)*, *scalloped (sd)*, *wingless (wg)*, *trachealess (trh)*, *forked (f)* and *tiptop (tio)*. It looked that these genes were very conservative in arthropods. The evolutionary trajectory for these genes' divergence requires further studies in the future so we did not explore this issue in the present manuscript.

Clark-Hachtel CM, Tomoyasu Y. Exploring the origin of insect wings from an evo-devo perspective. *Current opinion in insect science*, 2016 , 13:77–85

Line 302: 'with retinal...' is there a word missing?

Response:

Thanks for reviewer's suggestion. "necessary for the opsin activation" was missing. However, since we need to shorten the figure legend, this part has been removed.

Lines 314-337: this subsection is hard for to follow for the generalist reader. It is filled with terms that are not elaborated anywhere in the text (for example R&R1, D2, D4) and could be simplified. Additionally, the ecdysone cascade is presented with very little background and provides the reader with little information. For example, there is no reference to the ecdysone synthesis pathway, the Y organ, the fact that there are multiple nuclear receptors (including several ecdysone receptor isoforms), some of which are activated/inhibited by the ecdysone peak. It calls for a more elaborated

and structured writing or at least more references to reviews of these processes. Additionally, it is not clear how Br-C is linked to ecdysone. More details are required and should have been provided in the introduction as well.

Response:

Very good suggestions! In the revised ms, we added the related background on ecdysone synthesis pathway, the Y organ, nuclear receptors and Br-C *etc.* (Lines 321-337), with the addition of more references in this section.

Lines 554-564: this reads like introduction rather than discussion. It re-iterates some concepts that were introduced earlier and then provides new information that would suite better in the introduction.

Response:

Thanks for reviewer's suggestion. Since we have combined Results and Discussion, these sentences have been removed.

Reviewer #1 (Remarks to the Author):

The paper is much improved.

The addition of expression data regarding immune components adds a lot to the manuscript.

Minor comment:

The immunity section should reference some recent reviews of the crustacean immune system to orientate readers from outside the field of the diversity of immune molecules in crustaceans, as many are novel/unique to the crustaceans and arthropods.

Reviewer #2 (Remarks to the Author):

In my opinion, the authors have addressed all questions raised by the three reviewers adequately, and the manuscript is much improved.

I recommend acceptance without any further comments.

Responses to reviewers' comment:

Reviewer #1 (Remarks to the Author):

The paper is much improved.

The addition of expression data regarding immune components adds a lot to the manuscript.

Minor comment:

The immunity section should reference some recent reviews of the crustacean immune system to orientate readers from outside the field of the diversity of immune molecules in crustaceans, as many are novel/unique to the crustaceans and arthropods.

Response: We added a statement of crustacean immune system as shown below, and also some relevant reviews. "Crustaceans mainly rely on the innate immune system, consisting of humoral immunity and cellular immunity, to defend against pathogens (Li & Xiang, 2013)."

Reviewer #2 (Remarks to the Author):

In my opinion, the authors have addressed all questions raised by the three reviewers adequately, and the manuscript is much improved.

I recommend acceptance without any further comments.

Response: We thank the favorable comments from both reviewers.